# Seismic attenuation in Antarctic firn

**Stefano Picotti**[1], **José M. Carcione**[1,2], **and Mauro Pavan**[3]

[1]Section of Geophysics, National Institute of Oceanography and Applied Geophysics (OGS), Trieste, Italy
[2]School of Earth Sciences and Engineering, Hohai University, Nanjing, China
[3]Department of Earth, Environment and Life Sciences (DISTAV), University of Genoa, Genoa, Italy

**Correspondence:** Stefano Picotti (spicotti@ogs.it)

**Abstract.** We estimate the seismic attenuation of P and S waves in the polar firn and underlying ice by spectral analysis of diving, refracted, and reflected waves from active-source three-component seismic signals obtained in 2010 on the Whillans Ice Stream (WIS), a fast-flowing ice stream in West Antarctica. The resulting quality factors are then successfully modeled using a rock-physics theory of wave propagation that combines White's mesoscopic attenuation theory of interlayer flow with that of Biot/squirt flow. The first theory describes an equivalent viscoelastic medium consisting of a stack of two alternating thin porous layers, both of which have thicknesses that are much greater than the pore size but smaller than the wavelength. On the other hand, in the so-called Biot/squirt-flow model, there are two loss mechanisms, namely the global Biot flow and the local flow from fluid-filled microcracks (or grain contacts) to the pore space and back, where the former is dominant over the latter. The fluid saturating the pores is assumed to be fluidized snow, defined as a mixture of snow particles and air, such as powder, with a rigidity modulus of zero.

## 1 Introduction

The shallow parts of polar ice sheets and ice streams, commonly known as firn, are subjected to snow densification, a metamorphic process activated by the pressure gradient due to the accumulation of snow and by the temperature gradient at the near surface (Wilkinson, 1988). As a result, the density of firn increases continuously from the surface to the pore close-off depth, where the material has a density of about $840\,\mathrm{kg\,m^{-3}}$ [TS1] and can be considered glacial ice (Herron and Langway, 1980). Below this depth, the compression of englacial bubbles further increases the ice density until the maximum value is reached. In the cold and dry internal parts of the Antarctic continent, the densification is slow and the firn layer thickness exceeds $100\,\mathrm{m}$ (van den Broeke, 2008). On the other hand, in the high-strain shear margins of ice streams, this metamorphic process can be enhanced due to the effect of strain softening (Oraschewski and Grinsted, 2022), leading to exceptionally thin firn layers.

Because of its density structure, polar firn is generally laterally homogeneous, with the seismic velocity increasing nonlinearly with depth. This velocity profile results in the continuous refraction of seismic waves (Greenhalgh and King, 1981), which are also called diving waves. There are many examples in the literature that use diving waves to estimate the firn velocity–depth structure by picking and inverting first-break traveltimes (e.g., Kirchner and Bentley, 1979; King and Jarvis, 2007; Picotti et al., 2015). The ice-fabric characteristics as a function of depth have been obtained by exploiting the P- and S-wave anisotropic velocities inferred from active-seismic surveys conducted in different settings (e.g., Blankenship and Bentley, 1987; Picotti et al., 2015). However, while tomographic methods for estimating seismic velocity in the whole ice column are well established, algorithms for quantifying the depth dependence of attenuation are underdeveloped. In particular, as far as we know, there are no examples of the vertical profiling and modeling of the intrinsic seismic attenuation of P and S waves in the polar firn so far.

Intrinsic loss is often quantified using the inverse quality factor $1/Q$, which represents the fraction of wave energy lost to heat in each wave period (e.g., Carcione, 2022; Gurevich and Carcione, 2022). P-wave quality factors ($Q_P$) in ice have been measured by several authors in various depth ranges.

They were found to range from as low as 5 in the temperate ice at the surfaces of mountain glaciers (Gusmeroli et al., 2010) up to 700 within cold polar ice caps (Bentley and Kohnen, 1976). This wide range of values indicates a strong dependence of the quality factor on temperature, as demonstrated by laboratory experiments (Kuroiwa, 1964). This dependence was also verified by Peters et al. (2012) from seismic measurements in Greenland, where $Q_P$ decreases with depth due to an increase in temperature. In this case, $Q_P$ was measured within narrow depth ranges using strong basal and englacial reflections. Furthermore, it is common practice to measure the average $Q_P$ over the entire ice column using the primary and multiple reflection events (e.g., Holland and Anandakrishnan, 2009; Booth et al., 2012). Regarding the S-wave quality factor ($Q_S$), Clee et al. (1969) and Carcione et al. (2021) measured $Q_S$ in warm mountain glacier ice and reported values of about 23 and 12, respectively. To our knowledge, no other measurements of $Q_S$ in glacial ice or firn have been published in the literature.

While cold polar ice attenuation is low, this is not the case for polar firn. Bentley and Kohnen (1976) obtained a value of the ice $Q_P$ from seismic refraction measurements at Byrd Station of about 715 at 136 Hz between 100 and 500 m depth and at an average temperature of about $-28\,°C$. Because this value is very high, the intrinsic attenuation in polar ice (below the firn) caused only a minor reduction in the signal amplitude in their case. On the other hand, Albert (1998) performed numerical simulations of seismic waves in firn at the South Pole, Antarctica, between 0 and 300 m, where he neglected seismic attenuation on the basis of Bentley and Kohnen's results. However, as we show in the present work, attenuation in firn cannot be neglected, since the $Q_P$ factor has values between 4 and 50 down to 40 m depth, thus inducing strong energy loss and velocity dispersion near the surface.

Regarding the attenuation mechanisms, Bentley and Kohnen (1976) assume that their results are "consistent with damping in slightly contaminated ice by a combination of two mechanisms: molecular relaxation at temperatures colder than about $-20\,°C$ and grain boundary viscosity at warmer temperatures". However, their study does not provide a physical predictive microstructural model for attenuation. Grain boundary relaxation and melting theories based on the Arrhenius equation can be important at temperatures close to the melting point of ice. In the case of rocks, such a model has been applied to describe seismic attenuation at the crust and mantle (Carcione et al., 2018).

Here we consider three attenuation mechanisms based on physical models described in Pride et al. (2004), Carcione and Picotti (2006), and Carcione (2022). We show that wave-induced fluid flow generates enough heat to explain the levels of intrinsic attenuation. This flow occurs at different spatial scales that can broadly be categorized as "macroscopic", "mesoscopic", and "microscopic". The macroscopic flow is the wavelength-scale equilibration that occurs between the

peaks and troughs of a P wave. This mechanism was first treated by Biot (1956) and is simply referred to as "Biot loss". The microscopic flow occurs when a wave squeezes a porous medium that has grain contacts and cracks; these respond with a greater fluid pressure than the main pore space, resulting in a flow from cracks to pores named "squirt flow". Finally, mesoscopic length scales are those much larger than grain sizes but smaller than wavelengths. Heterogeneity across these scales may be due to frame variations or to patches of different immiscible fluids. In our case, because of the seasonal alternation, the firn can be considered a finely layered medium created through the deposition of two porous media, with one layer being snow-like with high porosity and the other ice-like with low porosity (e.g., van den Broeke, 2008). When a compressional wave squeezes such a medium, the effect is similar to squirt flow, with the more compliant portions of the material responding with a greater fluid pressure than the stiffer portions. What follows is a flow of fluid capable of generating a significant loss in the seismic band. In the case of Antarctic firn, the seismic wave attenuation cannot be explained by adopting a simple porous model consisting of a rigid structure of ice and a porous space filled with air. Such a simple ice–air model underestimates the seismic attenuation by orders of magnitude. Therefore, the fluid saturating the pore space in this case is assumed to be fluidized snow, a mixture of snow particles and air (Mellor, 1974; Maeno and Nishimura, 1979; Nishimura, 1996). In this study, we demonstrate that the replacement of air with pore-fluidized snow leads to increased attenuation and to quality factor values comparable with those obtained from seismic data. To our knowledge, this is the first attempt to use the concept of fluidized snow as a porous fluid in Biot's theory to model the wave attenuation in firn.

In the present work, we infer the P- and S-wave quality factor vertical profiles of the firn layer of Whillans Ice Stream (WIS – West Antarctica) using the three-component seismic data recorded on the Subglacial Lake Whillans (SLW) during the active-source experiment described in Horgan et al. (2012) and Picotti et al. (2015). First, we describe the data, the attenuation rock-physics theories, and the methodology used to extract the $Q$ factor from the seismograms. Then, we analyze the spectral content of the diving-wave first breaks and refracted waves at the firn bottom, and we compute the $Q_P$ and $Q_S$ factors using the frequency-shift technique. Finally, we fit the two vertical $Q$ profiles using the presented rock-physics theories. In the "Discussion", we give a detailed explanation of the importance of this model in studies related to the physical properties of the firn and to the characterization of subglacial media using amplitude variations with offset (AVO) analysis (e.g., Peters et al., 2008; Booth et al., 2012). We describe an alternative procedure for calculating the average $Q_P$ and $Q_S$ of the ice column below the firn that uses the reflected waves at the bottom of the glacier and the $Q$ profile of the firn. This procedure can be useful in cases

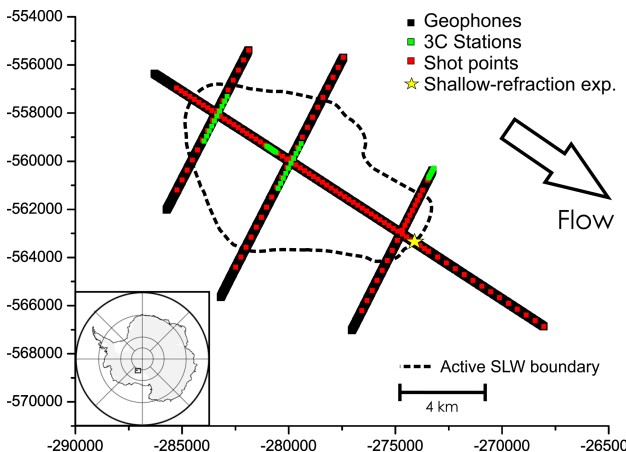

**Figure 1.** Map showing the location of SLW and the seismic survey geometry (modified from Picotti et al., 2015). The location of the shallow-refraction experiment is indicated. Polar stereographic projection with true scale at $-71°$.

where conventional methods of amplitude preconditioning for AVO analysis are not applicable.

## 2 Seismic data

In the austral summer of 2010–2011, a comprehensive surface geophysics program surveyed a location on the WIS. The main target of the survey (Horgan et al., 2012; Christianson et al., 2012) was the seismic and radar characterization of the SLW (Fig. 1), an active subglacial lake that is the subject of a subglacial access program Tulaczyk et al. (2014). Four active-source seismic lines were acquired during this program: one line parallel to the ice stream flow direction, following the long axis of the lake, and three transverse lines across the lake (Fig. 1).

Two types of seismic surveys were undertaken: one to obtain the characteristics of the firn and another for the imaging (Horgan et al., 2012) and to define the anisotropy of the whole ice column (Picotti et al., 2015). For the analysis of the seismic velocity fields within the firn, a shallow-refraction dataset was generated using stacked hammer blows onto a wooden body embedded in the snow. This experiment was located on the longitudinal line, as shown in Fig. 1. The data were recorded on a 48-channel seismic system using georods consisting of five 40 Hz geophones wired in series (Voigt et al., 2013). The digital sample interval was 0.25 ms, and the maximum offset was 470 m, which is enough to define the velocity profile of the whole firn layer. To acquire the three components of the ground motion, the georods were rotated through the three mutually perpendicular axes. The horizontal longitudinal and transverse components were parallel and orthogonal to the line direction, respectively. In the shallow-refraction survey, the trace spacing was increased with increasing offset to gain better resolution in the shallow part of

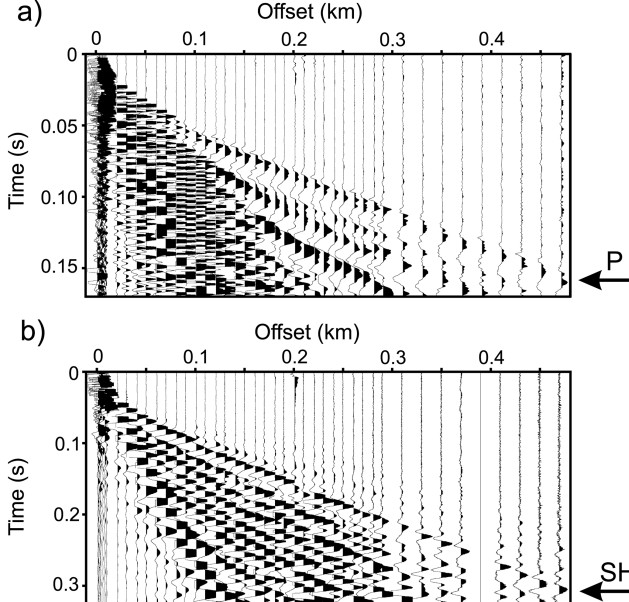

**Figure 2.** Hammer shallow-refraction seismograms recorded by the georods along the longitudinal line (modified from Picotti et al., 2015). The vertical component is shown in **(a)**, while the horizontal transverse component is shown in **(b)**. To acquire the two components, the georods were rotated along the corresponding mutually perpendicular axes. The P and SH diving-wave first breaks are indicated.

the firn, which exhibits the strongest velocity gradient. The trace spacing was 1, 10, and 20 m for offsets lower than 10 m, those ranging between 10 and 290 m, and those larger than 290 m, respectively. For the purposes of this work, the generation and acquisition of S waves polarized in the orthogonal direction with respect to the seismic line (i.e., SH waves) is far better, because these are pure S waves, free of P-wave interferences. We also acquired the SV waves, which were polarized in the longitudinal vertical plane, but these signals are more affected by interferences from diving and multiply refracted P waves propagating in firn. The obtained seismograms are shown in Fig. 2, where the P and SH diving-wave first breaks picked for the traveltime inversion and for the spectral analysis are indicated.

All the other data (Fig. 1), which were used to perform the imaging and to define the seismic anisotropy of the entire ice column, were generated using 0.4 kg PETN (pentaerythritol tetranitrate) charges placed at a depth of 27 m by using a hot water drill. Data were recorded on two 48-channel seismic systems, and the sensors consisted of alternating single vertical 28 Hz geophones and georods spaced 20 m apart (Fig. 3a). In addition, multi-component data were acquired using three-component (3C) continuous recording stations (Fig. 3b). The 3C stations were first placed along the longitudinal line and spaced 24 m apart, after which they were moved to the transverse lines, where the distance was

24 or 240 m (Fig. 1). Each of these stations consisted of a Guralp 40-T broadband seismometer with a 40 s corner period that was coupled to a Reftek RT-130 data acquisition system equipped with GPS timing. The maximum offset was 4320 m for the data recorded using 3C stations and 1910 m for the other data. These data are described in more detail by Horgan et al. (2012) and Picotti et al. (2015).

## 3 Theory and methodology

### 3.1 Overview of the theory

The theory of wave propagation in firn combines White's mesoscopic attenuation theory of interlayer flow (White et al., 1975; Carcione and Picotti, 2006; Carcione, 2022) and that of Biot/squirt flow (Gurevich et al., 2010; Carcione and Gurevich, 2011; Carcione, 2022). The first theory describes an equivalent viscoelastic medium consisting of a stack of two thin alternating porous layers of thickness $d_1$ and $d_2$, such that the period of the stratification is $d = d_1 + d_2$. The theory gives the complex and frequency-dependent stiffness $E$, equivalent to the P-wave modulus. The complex velocity is given in Appendix A. On the other hand, in the so-called Biot/squirt-flow model, there are two loss mechanisms, namely the Biot global-flow one (Biot, 1956) and local flow from fluid-filled microcracks (or grain contacts) to the pore space and back. Figure 4 shows a sketch of two sandstone grains in contact.

Gurevich et al. (2010) assumed that a compliant pore forms a disk-shaped gap between two grains and its edge opens into a toroidal stiff pore, where $R$ is the radius of the contact (or crack) and $h$ is its thickness. The model assumes that the material becomes stiffer when the fluid pressure does not have enough time to equilibrate between the stiff and compliant pores (grain contacts and main voids, respectively). The complex velocity is given in Appendix B.

### 3.2 Phase velocity and quality factor

Denoting the complex velocities corresponding to the mesoscopic (subscript 1) and Biot/squirt-flow (subscript 2) loss models as $v_1$, $v_{2P}$, and $v_{2S}$, the global P-wave phase velocity and dissipation factor are

$$v_{pP} \approx v_{p2P}, \tag{1}$$

and

$$\frac{1}{Q_P} = \frac{1}{Q_1} + \frac{1}{Q_{2P}}, \tag{2}$$

respectively, where

$$v_{p2P} = \left[ \text{Re} \left( \frac{1}{v_{2P}} \right) \right]^{-1}, \tag{3}$$

and

$$Q_1 = \frac{\text{Re}(v_1^2)}{\text{Im}(v_1^2)}, \quad Q_{2P} = \frac{\text{Re}(v_{2P}^2)}{\text{Im}(v_{2P}^2)}. \tag{4}$$

In the above equations, Re and Im denote the real and imaginary parts. Since the dominant mechanism is the Biot/squirt-flow one, we assume as a first approximation that the P-wave phase velocity is that of this loss mechanism. Alternatively, the phase velocity (Eq. 1), $v_{pP}(\omega)$, can be obtained from $Q_P(\omega)$ by using the Kramers–Kronig relations as given in Eq. (8) of Carcione et al. (2020) (see also Eq. 2.141 of Carcione, 2022).

Equation (2) can be demonstrated if we consider that the decay factor of a plane wave along a distance $r$ due to the effect of the two attenuation mechanisms is

$$\exp(-\alpha_1 r) \exp(-\alpha_{2P} r), \tag{5}$$

where the $\alpha_1$ and $\alpha_{2P}$ terms are attenuation factors given by

$$\alpha_1 = \frac{\pi f}{v_{p1} Q_1}, \quad \alpha_{2P} = \frac{\pi f}{v_{p2P} Q_{2P}}. \tag{6}$$

For low-loss media, $v_{p1}$ is the mesoscopic-loss phase velocity and $f$ is the frequency (see Eq. 2.129 in Carcione, 2022). Substituting Eq. (6) into Eq. (5), we obtain

$$\exp \left( -\frac{\pi f r}{v_{pP} Q_P} \right), \tag{7}$$

where we have assumed that $v_{p1} \approx v_{p2P}$.

On the other hand, the properties of the S wave are solely described by the Biot/squirt-flow model, such that

$$v_{pS} = \left[ \text{Re} \left( \frac{1}{v_{2S}} \right) \right]^{-1}, \tag{8}$$

and

$$Q_S = \frac{\text{Re}(v_{2S}^2)}{\text{Im}(v_{2S}^2)}. \tag{9}$$

### 3.3 Estimation of the quality factor

To estimate the quality factor from the seismograms, we adopted the frequency-shift method (e.g., Quan and Harris, 1997; Picotti and Carcione, 2006; Picotti et al., 2007). Let $S(f)$ and $R(f)$ be the quasi-Gaussian spectra observed at the source and at a receiver, respectively, located at a mutual distance $d$ in a homogeneous and isotropic medium. The frequency-shift approach is based on the property that, as the wave propagates through the medium, the high-frequency part of the spectrum decreases faster than the low-frequency part (Quan and Harris, 1997). This effect may be quantified by measuring the resulting downshift $\Delta f = f_S - f_R$, where

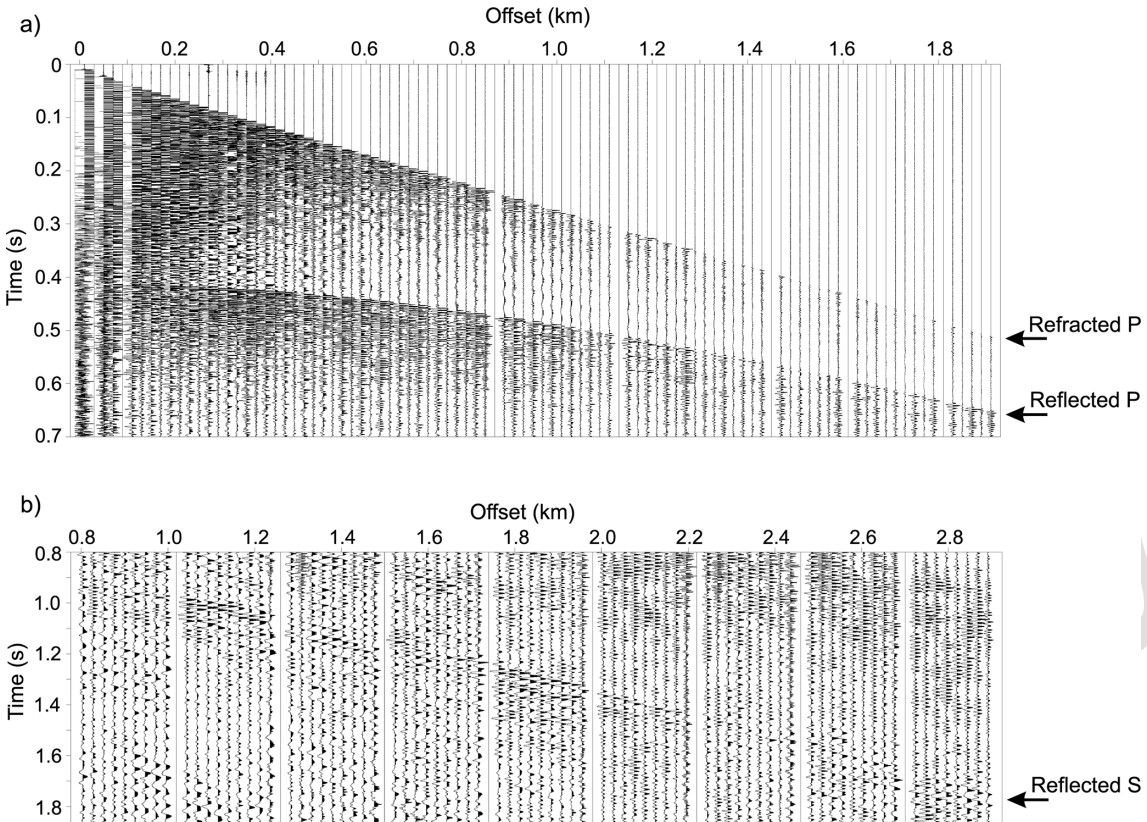

**Figure 3.** Explosive-charge seismograms recorded by the vertically oriented geophones and georods **(a)** and by the 3C stations on the horizontal transverse component (modified from Picotti et al., 2015) **(b)**. A 10–400 Hz band-pass filter was applied to remove the surface waves. The P and S waves reflected at the ice–sediments interface, as well as the P waves refracted at the bottom of the firn layer, are indicated.

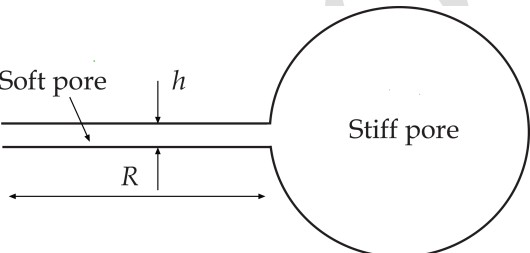

**Figure 4.** Sketch of the squirt-flow model, where two sandstone grains in contact are shown. The soft pores are the grain contacts and the stiff pores constitute the main porosity. The quantity $R$ is the radius of the disk-shaped soft pore (half a disk is represented in the plot).

$f_S$ and $f_R$ are the frequency centroids of $S(f)$ and $R(f)$, respectively. Then, if we approximate the spectrum $S(f)$ by a Gaussian with variance $\sigma_S^2$, we have

$$Q = \frac{\pi d \sigma_S^2}{v \Delta f} = \frac{\pi \Delta t \sigma_S^2}{\Delta f}, \tag{10}$$

where $v$ is the P- or S-wave velocity and $\Delta t$ is the traveltime. Although this method is only applicable under the hypothesis of constant $Q$ in the bandwidth of interest (Quan and Harris, 1997; Dasgupta and Clark, 1998), it can also be applied with good reliability to frequency-dependent $Q$ media. For example, Picotti et al. (2007) estimated the frequency-dependent $Q$ from numerically simulated seismograms by using the spectral-ratio and frequency-shift methods, which gave values in good agreement with the synthetic theoretical models. Moreover, they found that the classical spectral-ratio method is more sensitive to this problem, resulting in noticeable departures of the logarithm of the spectral ratio from the linear trend. On the other hand, the frequency-shift method is less sensitive to this problem when the spectrum of the signal is Gaussian because the quality factor variation is generally weak in the frequency band of active seismic experiments (typically 5–500 Hz). It becomes important only towards the edges of the seismic frequency band, where the integral contributions of the spectral amplitudes to the calculation of the variances and the frequency centroids are negligible (e.g., Picotti and Carcione, 2006).

In the more general case of inhomogeneous media, the frequency shift along a raypath can be expressed as

$$\Delta f = \sigma_S^2 \int\limits_{\text{ray}} \alpha_0 \mathrm{d}l = \pi \sigma_S^2 \int\limits_{\text{ray}} \frac{\mathrm{d}l}{Qv}, \qquad (11)$$

Quan and Harris (1997), where the integral is taken along
the raypath and $\alpha_0 = \pi / Qv$ is the attenuation coefficient,
i.e., the attenuation factor is linearly proportional to the frequency and defined as $\alpha_0 f$. To infer the vertical $Q$ profile
of firn, we consider the whole firn column as a sequence of
homogeneous-$Q$ layers. The $Q$ of each individual layer is
obtained by adopting a layer-stripping method that considers the cumulative attenuation of all the overlying layers and
exploits the monotonic increases in the firn's P- and S-wave
velocities. The layer-stripping method is a well-established
technique in both traveltime and attenuation tomography
(e.g., Yilmaz, 2001; Böhm et al., 2006; Rossi et al., 2007).
The P- and S-wave velocity fields (Fig. 5a) were previously
computed in Picotti et al. (2015) by the traveltime inversion of first arrivals, following the method of Herglotz and
Wiechert (Herglotz, 1907; Wiechert, 1910; Nowack, 1990).
This technique, applied for the first time to Antarctic seismic
data by Kirchner and Bentley (1979), calculates the velocity–
depth function from traveltime-offset-picked first breaks. It is
well suited to situations where the velocity increases monotonically with depth, and for this reason it was successfully
applied to firn by several authors (e.g., King and Jarvis,
2007). The resulting diving-wave paths in the velocity gradient (Fig. 5b) are obtained using an optimized ray-tracing
algorithm based on the shooting method, as described in Picotti et al. (2015). The curves displayed in Fig. 5a represent
improved versions of those originally presented in Picotti et
al. (2015).

Let's consider layer $N$ in this succession and assume that
the value of $Q$ in the overlying $N-1$ layers is already known.
In our plane-layered model, the depth of each layer coincides
with the depth of maximum penetration of each ray at increasing offsets. The ray corresponding to this layer reaches
a receiver at the surface whose recorded first-break frequency
centroid is $f_R$. Under our hypothesis, and in accordance with
Eq. (11), the total frequency shift in the overlying $N-1$ layers is

$$\Delta f_{N-1} = 2\pi \sigma_S^2 \sum_{i=1}^{N-1} \frac{d_i}{Q_i v_i}, \qquad (12)$$

where $v_i$ and $d_i$ are the velocities and the lengths of the ray
segments in each layer, respectively, and the factor of 2 accounts for both the downgoing and the upgoing waves. Then,
the quality factor of the layer $N$ is obtained by Eq. (10),
where $d = d_N$, $v = v_N$, and the frequency shift in the layer
$N$ is given by $\Delta f = \Delta f_N = f_S - f_R - \Delta f_{N-1}$.

The layer-stripping method requires that the quality factor of the shallowest layer is known a priori. Moreover, the

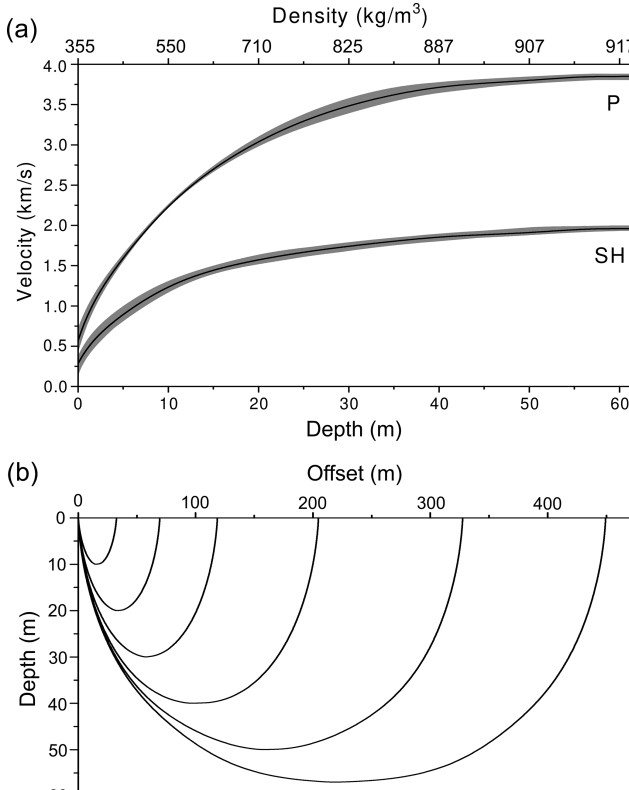

**Figure 5.** P- and S-wave velocity profiles as a function of depth obtained by using the Herglotz–Wiechert traveltime inversion method **(a)**, where the error in velocity is displayed as gray bands, and P-wave diving paths corresponding to some picked first breaks **(b)**. These plots are improved versions of those originally presented in Picotti et al. (2015). The velocity curves as a function of density, obtained using Eq. (13) proposed by Kohnen (1972), are also presented.

frequency centroid $f_S$ and variance $\sigma_S^2$ of the source must
be known as well. These two values can be estimated using
the first breaks recorded by the receivers placed close to the
source. For this purpose, it is necessary that the acquisition
geometry is properly designed to have a sufficient number of
traces at very short offsets. Then, particular care must be devoted to acquiring first breaks with high signal quality, e.g.,
the wavelets must be free of interference effects and their
amplitudes must not be clipped. If such signals are available,
the following considerations allow for the computation of the
required spectral properties. It is convenient in this case to
estimate the difference in the frequency content of propagating wavelets at different receivers. We consider the closest
available signal to the source as reference wavelet, which is
compared with the first breaks recorded at increasing offsets.
In other words, we compare the spectrum $R_1(f)$ of the reference wavelet to that of another wavelet $R_2(f)$, which propagates for an additional traveltime $\Delta t$ in the same shallow
layer. If the traces are sufficiently close to each other and to

the source, it is possible to characterize the quality factor of the uppermost layer as the average of all calculated values, and the variance $\sigma_S^2$ can be approximated to that of the reference wavelet. Finally, the dominant frequency of the source $f_S$ follows from Eq. (10) by using the reference first-break traveltime.

## 4 Results

### 4.1 Quality factor vertical profiling

As previously described, a requirement for the layer-stripping technique is knowledge of the quality factor of the shallowest layer as well as the dominant frequency and variance of the source. We determined these properties by using the first breaks recorded close to the source: up to a maximum offset of 10 and 6 m, for P and S waves, respectively.

Figure 6 shows the time histories and the corresponding spectra of the considered P-wave first breaks.

Unfortunately, the first breaks recorded at offsets of less than 7 m are contaminated by the SV waves, and the amplitudes very close to the source are clipped. Taking the signal recorded at 7 m offset as the reference wavelet, we computed the frequency shifts and, using Eq. (10), we estimated the average P-wave quality factor (and standard deviation) of the shallowest layer to be $Q_P = 4.1 \pm 1.3$. Calculating the maximum penetration depth of the ray emerging at 10 m offset, we estimated the thickness of this layer to be about 2.7 m. Moreover, the estimated dominant frequency and variance of the source are $f_S = 574$ Hz and $\sigma_S = 171$ Hz, respectively.

Figure 7 shows the wavelets and the corresponding spectra of the considered S-wave first breaks.

Because the amplitudes of the first break recorded at 1 m offset are clipped, this trace cannot be used as the reference. Taking the signal recorded at 2 m offset as the reference wavelet, we computed the frequency shifts and, using Eq. (10), we estimated the average S-wave quality factor (and standard deviation) of the shallowest layer to be $Q_S = 1.9 \pm 0.4$. By calculating the maximum penetration depth of the emerging ray at an offset of 6 m, we estimated the thickness of this layer to be about 1.6 m. Furthermore, the estimated dominant frequency and variance of the source are $f_S = 496$ and $\sigma_S = 146$ Hz, respectively.

The frequency centroids of the diving P- and S-wave first breaks are obtained from the spectra of the selected waveforms, as described in Picotti and Carcione (2006). The amplitude integrals are calculated, for each wavelet, in the frequency band from zero to the maximum frequency of the signal. Since this high cutoff frequency depends on the signal-to-noise ratio, a statistic is calculated to evaluate the dispersion due to noise. The mean values of the obtained distributions of the frequency centroids for both the P- and S-wave first breaks are shown in Fig. 8. The corresponding standard deviations are less than 3 Hz. Both curves show a rapid decrease in the dominant frequency of the signal as a function of offset, except for an increase in frequency for the P waves between 10 and 30 m offset, which is an effect due to the diving waves traveling in that strong velocity–$Q$ gradient.

As described above, the P- and S-wave velocity fields shown in Fig. 5a represent improved versions of those originally computed by Picotti et al. (2015) by the traveltime inversion of first arrivals, following the traveltime method of Herglotz and Wiechert CE1 . The uncertainties in the velocities were obtained by perturbing the first-break traveltimes according to the dominant frequency of the selected wavelets and then repeating the Herglotz–Wiechert inversion to produce different velocity profiles as a function of depth. The averages and standard deviations of the obtained velocity distributions are displayed in Fig. 5a. The maximum P- and S-wave velocities, verified using the first arrivals refracted at distant offsets on the seismograms acquired using the explosive source, are $3864 \pm 30$ and $1947 \pm 25$ m s$^{-1}$ at $60 \pm 5$ m depth, respectively. At short offsets, errors increase due to the steep velocity gradient near the surface.

The two frequency curves shown in Fig. 8, the velocity profiles, the characteristics of the spectral source, and the $Q_P$ and $Q_S$ of the most superficial layer are the inputs for the layer-stripping procedure used to calculate the P- and S-wave quality factor profiles of the entire firn column. The uncertainties in the $Q_P$ and $Q_S$ values are obtained by repeating the procedure using the previously calculated frequency centroid and velocity distributions and perturbing the quality factors of the most superficial layers by their standard deviations. Then, from the two $Q$-factor distributions in the output, we derive the corresponding mean and standard deviation values with respect to depth.

Figures 9 and 10 display the resulting $Q_P$ and $Q_S$ profiles for the whole firn column, from the surface to the ice, together with the dominant frequency computed at the maximum penetration depth of each diving raypath.

The first plot (Fig. 9) shows a slow increase in $Q_P$ from the previously computed minimum value of $4.1 \pm 1.3$ close to the surface to a value of $120 \pm 35$ at about 50 m depth. Then, it increases sharply to over $300 \pm 140$ at about 58 m depth. A noticeable increase in uncertainty below 40 m depth is due to a consistent reduction in both the traveltime and the frequency shift in each layer. The second plot (Fig. 10) shows a moderate increase in $Q_S$ from the previously calculated minimum value of $1.9 \pm 0.4$ near the surface to an average maximum value of about $250 \pm 90$, which remains almost constant at depths greater than about 40 m.

Since the offset range is not sufficient to appreciate where the $Q_P$ factor stabilizes at the maximum value in the ice, we use the dataset acquired with the explosive charges placed at 27 m depth (Fig. 3a) to analyze the refracted P-wave first breaks recorded at large distances up to the maximum available offset of 1910 m. Figure 11 shows an example of selected wavelets and the corresponding spectra together with a reference wavelet recorded at 670 m offset. All these sig-

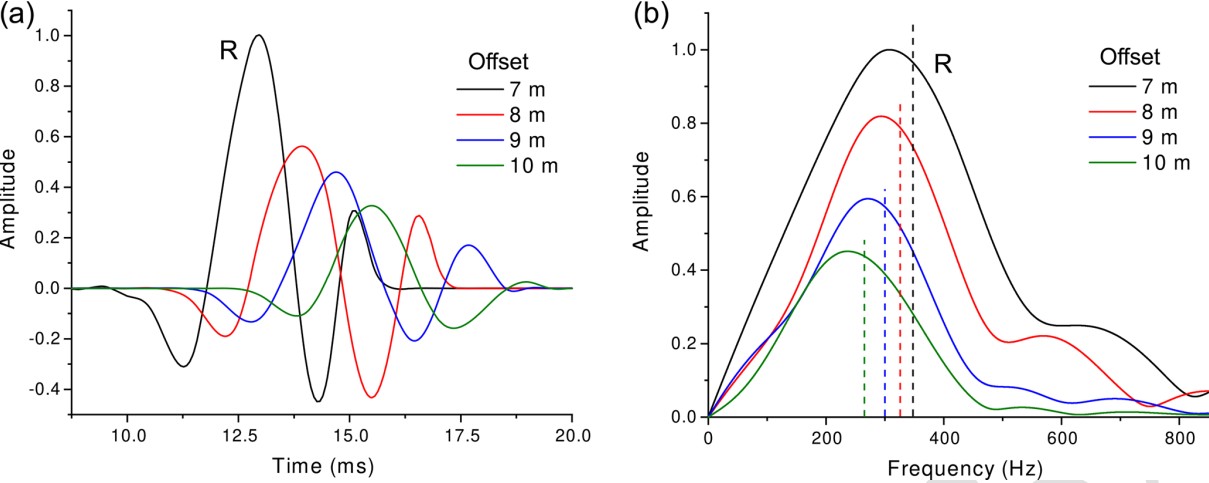

**Figure 6.** Time histories **(a)** and corresponding spectra **(b)** of the P-wave first breaks recorded between 7 and 10 m. The label *R* indicates the reference signal recorded at 7 m offset, while the vertical dashed lines indicate the frequency centroids.

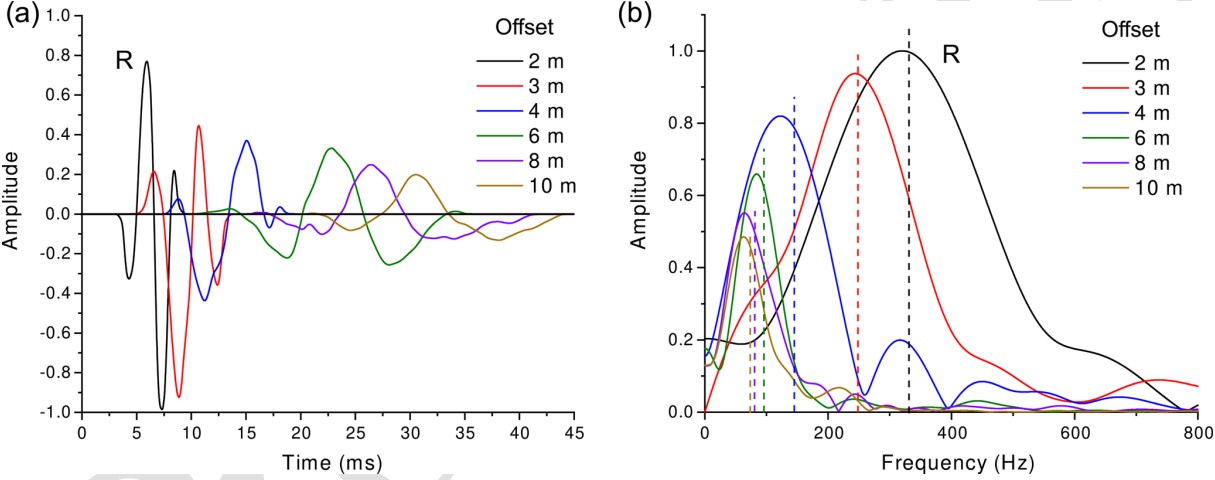

**Figure 7.** Time histories **(a)** and corresponding spectra **(b)** of the S-wave first breaks recorded between 2 and 10 m. The label *R* indicates the reference signal recorded at 2 m offset, while the vertical dashed lines indicate the frequency centroids.

nals travel along similar raypaths in the firn, while their paths in the underlying ice have different lengths. Following the same principle adopted for the computation of the $Q$ factor in the shallowest layer, we compare the spectrum of a reference wavelet to those of other wavelets that propagate for additional traveltimes in the deep ice. Because these paths through the ice are long, the differences in spectral centroid can be better appreciated and the quality factors can be calculated more reliably. This procedure should be repeated for a large number of selected wavelets in order to perform a statistical analysis. In this case, the average P-wave quality factor of the ice is $Q_P = 380 \pm 70$ at $60 \pm 5$ m depth, where the average measured temperature is about $-24\,°C$ (Engelhardt and Kamb, 1993). Using the same procedure, we also analyzed the refracted S-wave first breaks from 330 to 470 m offset (Figs. 2b and 8), resulting in an average ice S-wave qual-

ity factor of $Q_S = 260 \pm 80$ at $60 \pm 5$ m depth. This value confirms the result provided by the analysis of the diving waves.

## 4.2 Modeling of the seismic properties of the firn

Firn is assumed to be a deposition of two porous media, with one layer being snow-like with high porosity and the other ice-like with low porosity. The grains (ice) have the properties $K_s = 10$ GPa, $\mu_s = 5$ GPa (shear modulus), and $\rho_s = 917\,\mathrm{kg\,m^{-3}}$ in both layers (Gurevich and Carcione, 2022). The fluid saturating the pores is assumed to be fluidized snow, which is defined as a mixture of snow particles and air, like powder, with a rigidity modulus of zero. We consider $K_f = 571$ MPa, $\rho_f = 200\,\mathrm{kg\,m^{-3}}$ and $\eta = 0.1$ Pa s TS3. The properties of this material have been investigated by

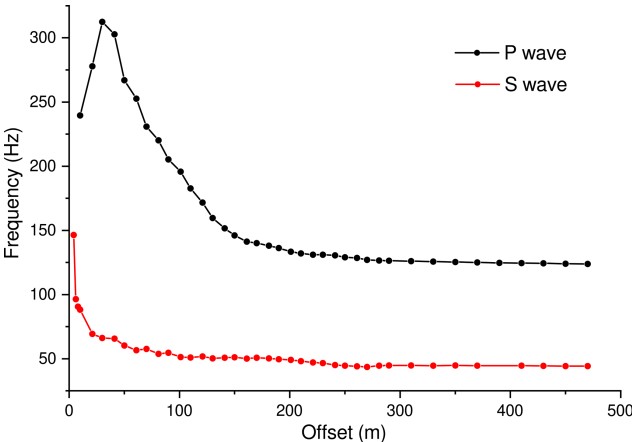

**Figure 8.** Dominant frequency of the diving P-wave **(a)** and S-wave **(b)** first break as a function of offset.

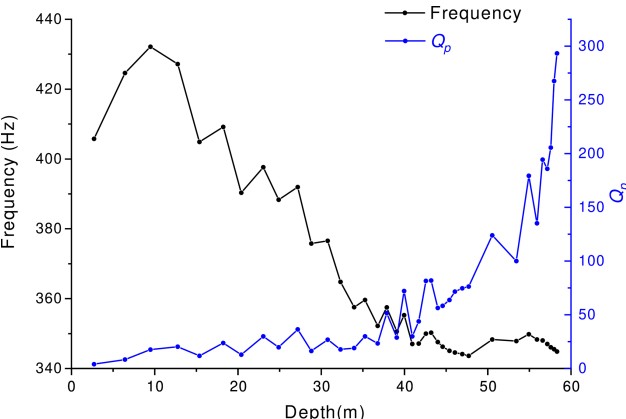

**Figure 9.** P-wave quality factor and dominant frequency as a function of depth, obtained from the layer-stripping frequency-shift method.

Mellor (1974), Maeno and Nishimura (1979), and Nishimura (1996).

For this study, it is important to measure the properties of the fluidized phase of the snow. This task can be performed during the coring of firn samples, as indicated in Nishimura (1991). The main apparatus consists of two parts: a fluidized snow feeder and an inclined chute, where it is possible to store the disintegrated snow in fluidized conditions. Measurements are made at a temperature of $-15\,^{\circ}\mathrm{C}$ to avoid adhesion effects between snow particles. In this context, the bulk density, elastic velocity, and viscosity can be measured.

The physical properties of the firn layer are obtained from the density model using functions of porosity that have been shown to be suitable for snow. The density profile as a function of depth is obtained by using the following empirical

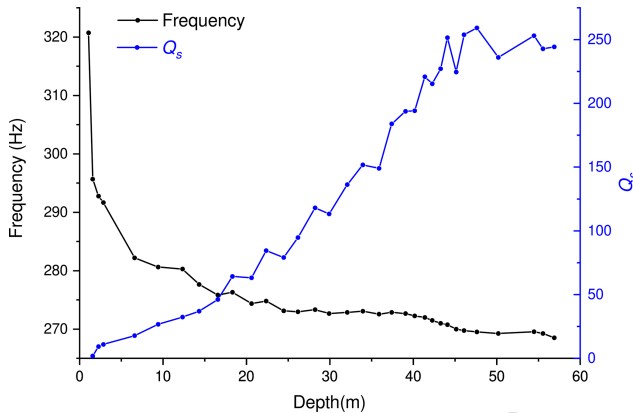

**Figure 10.** S-wave quality factor and dominant frequency as a function of depth, obtained from the layer-stripping frequency-shift method.

relationship (Kohnen, 1972):

$$\rho(z) = 0.917 \left[ 1 + \left( \frac{V_{\mathrm{P,ice}} - V_{\mathrm{P}}(z)}{2250} \right)^{1.22} \right]^{-1}, \quad (13)$$

where $V_{\mathrm{P}}(z)$ is the vertical P-wave velocity displayed in Fig. 5a, and $V_{\mathrm{P,ice}} = 3864\,\mathrm{m\,s^{-1}}$ is the velocity in ice, which we assume equal to the maximum computed P-wave velocity.

The porosity obtained from the experimental density (Eq. 13) is

$$\phi(z) = \frac{\rho_s - \rho(z)}{\rho_s - \rho_f}. \quad (14)$$

Figure 12 shows the experimental density and porosity, where the former increases and the latter decreases monotonically with depth, mainly due to compaction.

Then, for each layer, the dry-rock bulk modulus which best fits the data of Johnson (1982) is

$$K_m = K_s (1 - \phi)^{30.85/(7.76-\phi)}. \quad (15)$$

The dry-rock shear modulus is

$$\mu_{\mathrm{1S4}} = \frac{3}{2} \frac{1 - 2\nu}{1 + \nu} K_m, \quad \nu = 0.38 - 0.36\phi, \quad (16)$$

where $\nu$ is the Poisson ratio. Below the pore close-off depth, approximately 35 m in this case, the medium is mainly ice ($\rho \geq 840\,\mathrm{kg\,m^{-3}}$ and $\phi \leq 10\,\%$), and the Poisson ratio is better approximated from the inverted wave velocities as follows:

$$\nu = \frac{V_{\mathrm{P}}^2 - 2V_{\mathrm{S}}^2}{2\left(V_{\mathrm{P}}^2 - V_{\mathrm{S}}^2\right)} \approx 0.32, \quad (17)$$

(Mavko et al., 2009), where $V_{\mathrm{P}}$ and $V_{\mathrm{S}}$ are the P- and S-wave velocities displayed in Fig. 5a, respectively.

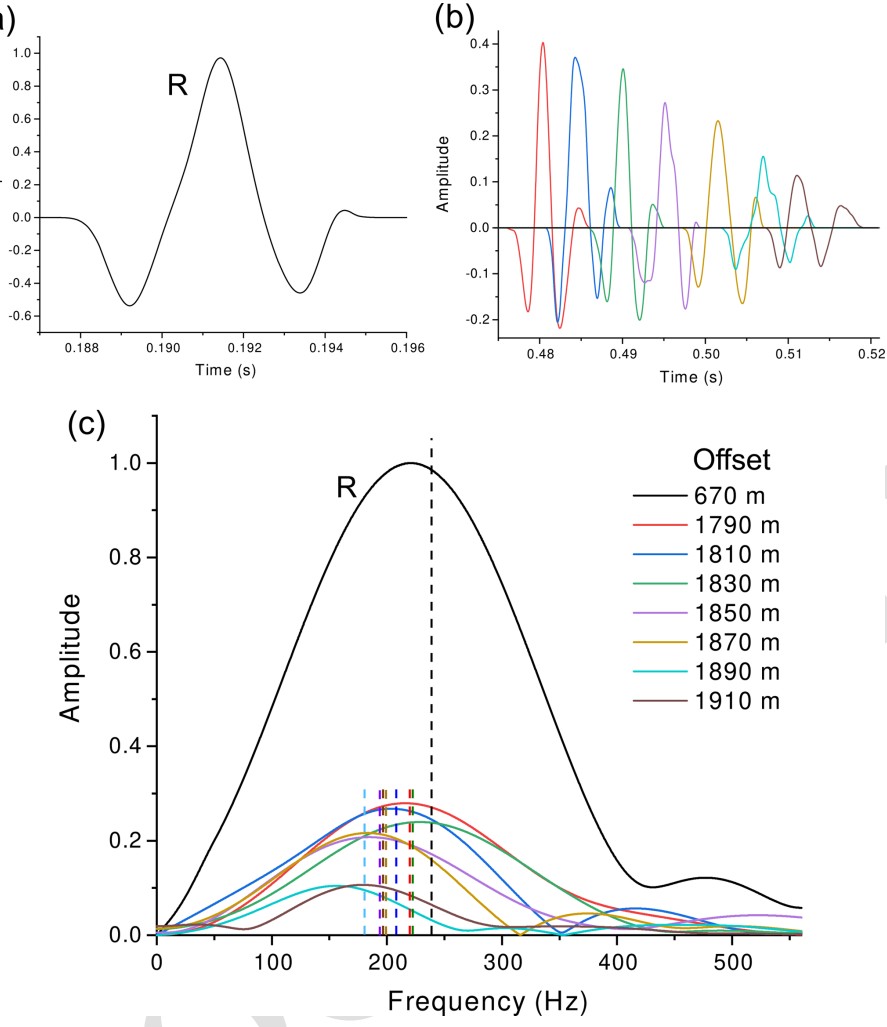

**Figure 11.** Time histories **(a, b)** and corresponding spectra **(c)** of the P-wave first breaks recorded at offsets of between 1790 TS2 and 1910 m. The label $R$ in **(a, c)** indicates the reference signal recorded at 670 m offset, while the vertical dashed lines in **(c)** indicate the frequency centroids.

The permeability is

$$\kappa = \frac{C}{\rho_s^2} \cdot \frac{\phi^3}{(1-\phi)^2}, \tag{18}$$

(Sidler, 2015; Gurevich and Carcione, 2022), where $C = 0.012\,\mathrm{kg}^2\,\mathrm{m}^{-4}$ and we consider that the porosity $\phi$ is obtained from the experimental velocities and density as given by Eqs. (13) and (14).

For the mesoscopic loss model, we assume that $\phi_1(z) \approx \phi(z)$, $p_1 = p_2 = 0.5$, and $\phi_2 = \gamma\phi_1$, where $\gamma$ is small, i.e., layer 2 (ice) has a much lower porosity than layer 1 (snow). Here we assume that $\gamma = 0.1$. On the other hand, the squirt-flow model has the following values of the parameters (Carcione and Gurevich, 2011): $h/R = 0.015$, $\phi_c = 0.0002$, and $K_h = 1.38K_m$, where $K_m$ is the bulk modulus with the grain contacts and cracks open. The attenuation is due to the softer layer with much higher porosity, since the cracks are open

and the global-flow loss (Biot) mechanism is effective. The contribution of the stiff layer is negligible.

Figure 13 shows the P- and S-wave velocities and dissipation factors as a function of frequency close to the surface, where $\phi_1 = 0.744$, $\phi_2 = 0.0744$ TS5, the solid and fluid properties of the two layers are the same. We can see that the dominant mechanism is the Biot (global flow) one (Biot, 1956), with a strong relaxation peak (high attenuation) and velocity dispersion. The squirt-flow peak is weaker and located at lower frequencies, while the mesoscopic loss contributes only to the P wave. The P- and S-wave dissipation peaks occur at a frequency of approximately 282 Hz. On the other hand, the model predicts that below a given depth (from 40 to 50 m), the squirt-flow loss predominates over the Biot global one.

Let us now study the dependence of attenuation on depth. We consider a frequency of 282 Hz, corresponding to the dis-

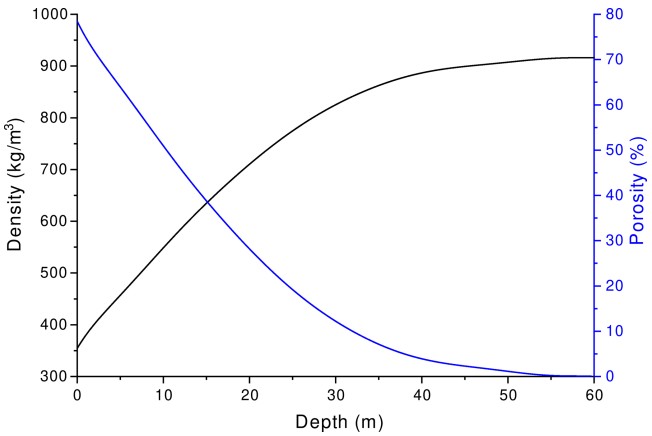

**Figure 12.** Density (black) and porosity (blue) as a function of depth.

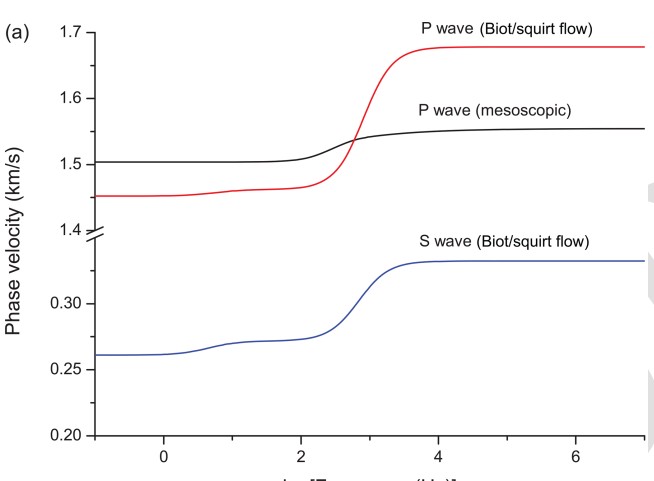

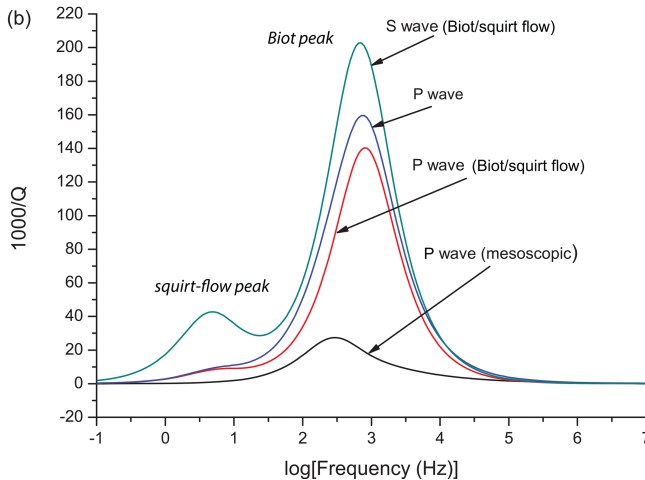

**Figure 13.** Analytical P-wave velocity **(a)** and dissipation factor **(b)** as a function of frequency close to the surface. TS6

sipation peaks in Fig. 13. Figure 14a shows a comparison between the experimental and theoretical quality factor profiles

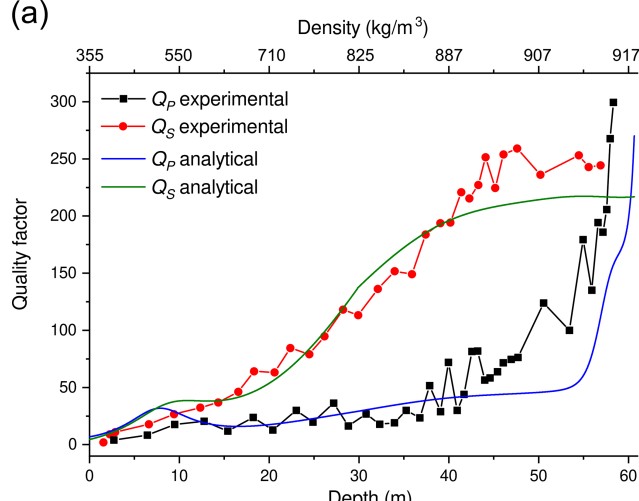

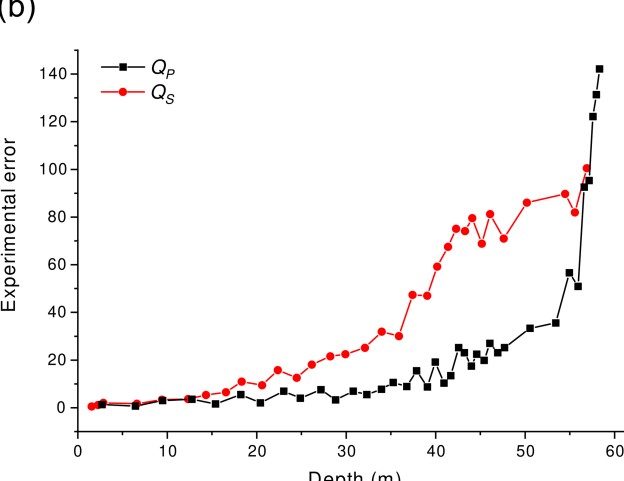

**Figure 14.** Comparison between the experimental (symbols) and theoretical (solid lines) P- and S-wave quality factors **(a)** as a function of depth. The quality factors are also represented as a function of density using Eq. (13) proposed by Kohnen (1972). Experimental errors **(b)** in the computation of $Q_P$ and $Q_S$ using the layer-stripping frequency-shift method.

as a function of depth, while Fig. 14b shows the experimental errors in $Q_P$ and $Q_S$. As expected, the uncertainties increase with increasing quality factor, in agreement with the fact that the attenuation effect weakens with increasing $Q$ and can be observed with high accuracy only over large distances. The differences between the theoretical quality factor and the profiled $Q$ are within the experimental errors for almost the entire firn column.

## 5   Discussion

Studying the structures of polar ice sheets and basal materials is essential for modeling the responses of ice masses to climate change. The mechanisms of the basal movement

of glaciers, which strongly influence the mass balance of Antarctica, are poorly understood. The presence at the ice bottom of enormous quantities of water-saturated sediments, ponds, and subglacial lakes determines the dynamics of the ice streams (e.g., Bindschadler et al., 2003; Winberry et al., 2009), which represent the main drainage conduits of ice from the interior of Antarctica to the ocean. Furthermore, the identification and characterization of subglacial lakes is also important for studying the climate history of Antarctica, including the possible presence of extremophile life (Siegert et al., 2011).

Seismic reflection techniques are powerful tools for mapping the physical properties of subglacial environments; specifically, the roughness at the subglacial boundary, water occurrence and fluid layer thickness, rock and sediment type, sediment porosity and fluid saturation, and the thickness of the subglacial sedimentary layer (e.g., Blankenship et al., 1987). These parameters are all crucial for understanding the basal flow mechanism, which is modulated by the subglacial hydrology and deformation characteristics of the subglacial till. Seismic imaging is the only way to map these properties over large areas, as drilling to the ice bottom is extremely expensive and only provides local information.

The application of advanced AVO techniques has successfully exploited detailed tomographic velocity models and reflection amplitudes at the basal boundary to gather considerable information on the underlying material properties in various contexts (e.g., Blankenship et al., 1987; Anandakrishnan, 2003; Peters et al., 2008; Booth et al., 2012). However, quantitative estimates of these subglacial properties are difficult due to the many challenges that characterize AVO techniques. The two most important limitations are that the source characteristics are often unknown and that the attenuation of P and S waves can be highly variable vertically across the entire ice column (and in some cases also laterally) and poorly constrained. Both of these conditions reduce the ability to quantify bedrock reflection coefficients, affecting uncertainties about the physical properties of subglacial materials.

Our method allows for a better understanding of the seismic properties of the firn, which are useful for estimating the ice-sheet mass balance from satellite observations of ice-sheet elevation (e.g., Wingham, 2000; Alley et al., 2007). As explained, since $Q$ is strongly influenced by the porosity (and density) profile of the propagation medium, measuring both velocities and quality factors has the potential to provide additional information on the physical structure of the firn, avoiding costly coring.

Furthermore, the present work provides alternative means to correct for seismic reflection amplitudes at the glacier floor and can be very useful in cases where conventional amplitude preconditioning methods for AVO analysis are impossible. Indeed, conventional methods ignore the complex $Q$ structure of the firn and usually rely on the presence of multiple reflections to calculate the source amplitude and average qual-

ity factor of the entire ice column at normal incidence (e.g., Holland and Anandakrishnan, 2009; Booth et al., 2012). In addition to the fact that multiple reflections are not always present in seismic data, when they are available, they often show a poor signal-to-noise ratio (e.g., Dow et al., 2013). In the case considered in this work, there is no evidence of multiple reflections in the overall dataset. Therefore, the obtained firn velocity–$Q$ profile can help in calculating both the source amplitude and the average quality factor of the ice mass below the firn.

Consider an explosive charge buried deep in the firn (in our case at a depth of about 27 m for most shots) and a ray propagating from the source to a surface receiver. Adopting our layered firn model, the source amplitude can be calculated considering the first breaks and Eq. (7), expressing the decay factor along a distance $r_i$ in each homogeneous layer. Neglecting the transmission loss between adjacent quasi-layers, and denoting the attenuation factor in layer $i$ by $\alpha_i = \pi f / Q_i v_i$, the total amplitude of the source $S(f)$ is

$$S(f) = GR(f) \prod_{i=1}^{N} \exp(\alpha_i \, r_i) = GR(f) A_c(f), \qquad (19)$$

where $A_c(f)$ is the amplitude correction factor, $f$ is the dominant frequency, $N$ is the number of crossed layers, $R(f)$ is the receiver amplitude, and $G$ takes into account frequency-independent factors, for example the geometric spreading factor, which is proportional to the total length of the ray. To reduce the uncertainty, Eq. (19) can be applied to a large number of receivers for each shot in order to perform a statistical analysis.

Let us now consider the amplitude $R(f)$ of a signal reflected from the bottom of the glacier. The reflection coefficient $R_c$ is obtained by correcting $R(f)$ for the decay factor as follows:

$$R_c(f) = \frac{GR(f) A_c(f)}{S(f)}, \qquad (20)$$

where the calculation of $A_c(f)$ is now performed along the rays reflected from the bed. Thus, this calculation implies knowledge of the quality factor profile of the entire ice column. In this case, in the absence of englacial and multiple reflections, the average quality factor between the firn bottom and the bed is required. This value can be estimated, for both P and S waves, by following the same principle adopted for the calculation of the $Q$ factor in the most superficial layer (direct waves at short offsets) and at the base of the firn (refracted waves at large offsets). This method does not require a priori knowledge of the source characteristics, only the firn's $Q$ profile. We compare the spectrum of a reference wavelet with those of other wavelets propagating for further traveltimes in both firn and deep ice. For this purpose, we select a reference wavelet at a short/medium offset (ray A) and another wavelet at a large offset (ray B), as shown in

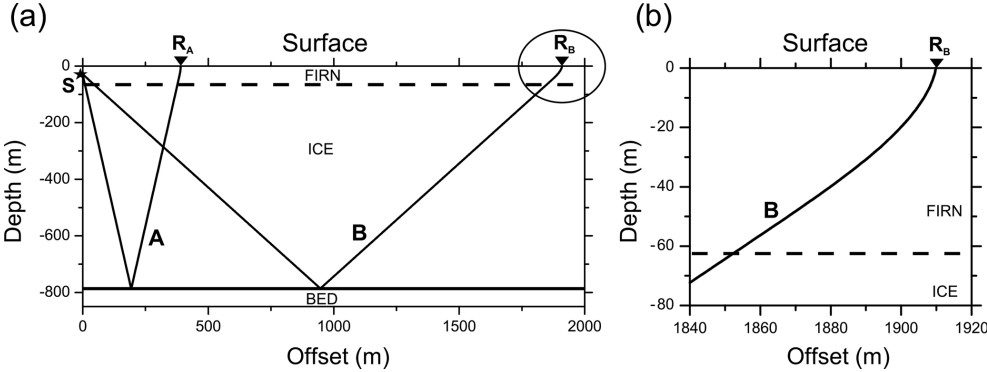

**Figure 15.** Ray tracing of P waves reflected from the bed and emerging at offsets of approximately 390 m (ray A – reference wavelet) and 1910 m (ray B), respectively **(a)**. Magnification of the P-wave ray emerging at the surface **(b)** in the area highlighted by the circle in **(a)**, showing the deflection caused by the velocity gradient in the firn. The source (S), an explosive charge placed at a depth of 27 m, and the receivers ($R_A$ and $R_B$) are indicated. The surface and ice-bottom interfaces are almost flat, with slopes of less than 1°.

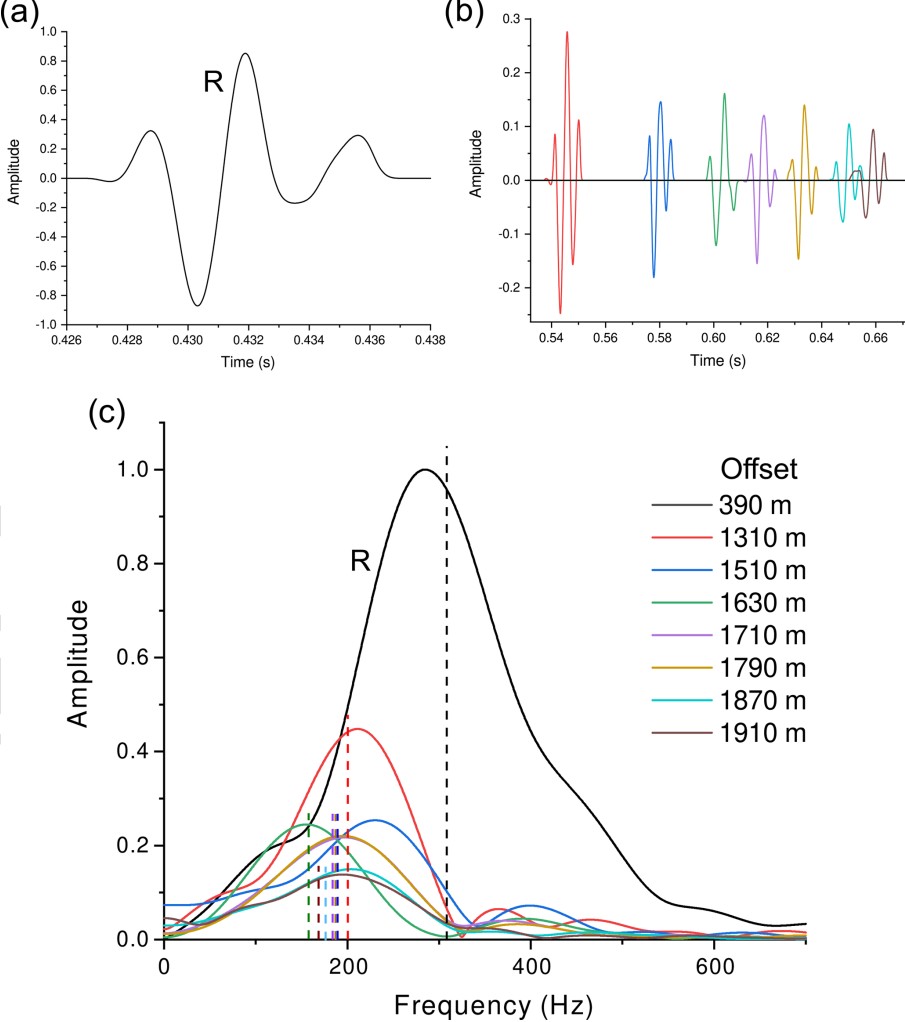

**Figure 16.** Time histories **(a, b)** and corresponding spectra **(c)** of the P-wave reflected wavelets recorded at offsets of between 1310 and 1910 m. The label $R$ indicates the reference signal **(a, c)** recorded at 390 m offset, while the vertical dashed lines indicate the frequency centroids.

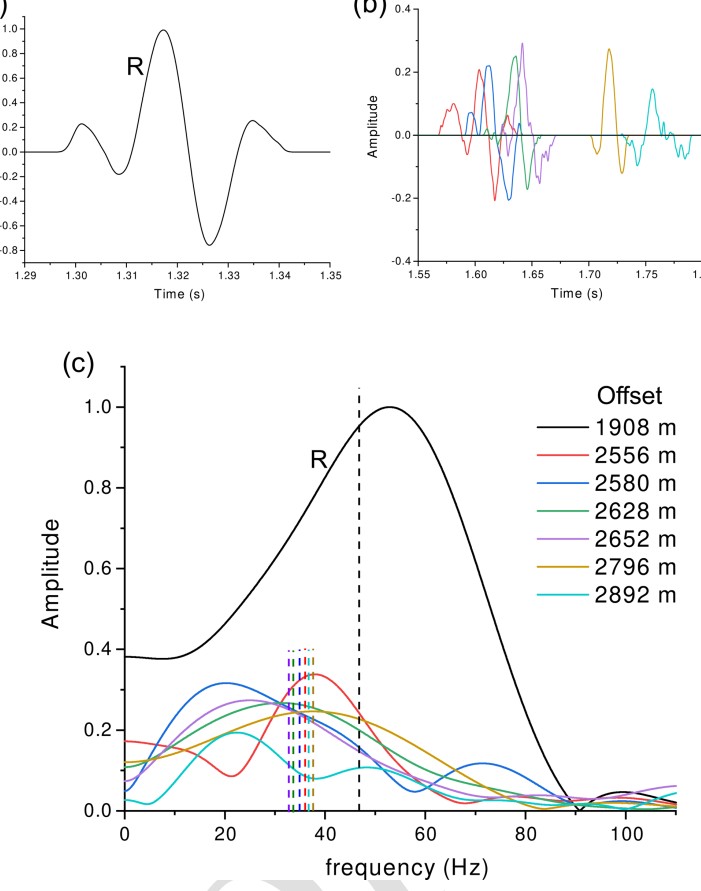

**Figure 17.** Time histories **(a, b)** and corresponding spectra **(c)** of the S-wave reflected wavelets recorded at offsets of between 1908 TS10 and 2892 m. The label $R$ indicates the reference signal **(a, c)** recorded at 1908 m offset, while the vertical dashed lines indicate the frequency centroids.

Fig. 15. These signals travel along raypaths with slightly dissimilar lengths in the firn, while their paths in the underlying ice have very different lengths. The greater the difference between the lengths of these paths, the larger the difference in the centroids of signal spectra and the greater the reliability of the calculated quality factor.

Applying Eq. (11) to rays A and B, we obtain

$$\sum_{\text{ray B}} \frac{\Delta t_k}{Q_k} - \sum_{\text{ray A}} \frac{\Delta t_k}{Q_k} = \frac{\Delta f_{BA}}{\pi \sigma_A^2}, \tag{21}$$

where $\Delta f_{BA}$ is the difference between the frequency centroids of the reflected wavelets recorded at the receivers $R_B$ and $R_A$ (Fig. 15). Separating the contributions of ice and firn in the summations, the average quality factor of the ice mass below the firn is

$$Q_{\text{ice}} = \frac{\sum_{\text{ray B}}^{\text{ice}} \Delta t_k - \sum_{\text{ray A}}^{\text{ice}} \Delta t_k}{\frac{\Delta f_{BA}}{\pi \sigma_A^2} - \left( \sum_{\text{ray B}}^{\text{firn}} \frac{\Delta t_k}{Q_k} - \sum_{\text{ray A}}^{\text{firn}} \frac{\Delta t_k}{Q_k} \right)}. \tag{22}$$

As done for the refracted waves at the bottom of the firn, we adopted the dataset acquired with the explosive charges

placed at a depth of 27 m (Fig. 3) in order to analyze the reflected waves from the glacier bottom. Figures 16 and 17 show two examples of selected wavelets and corresponding spectra up to maximum offset values of 1910 and 2892 m for P and S waves, respectively. These are the largest offsets available in our dataset that show signals with sufficient quality in terms of signal-to-noise ratio. We performed a ray tracing by adopting the algorithm described in Picotti et al. (2015), which allows the calculation of the ice velocity versus the propagation angle. Then, using the wavelet frequency centroids, the firn quality factor profiles, and the computed traveltimes along the ray segments, we obtained the average ice quality factors below the firn (for both P and S waves) using Eq. (22). This procedure should be repeated for a large number of selected wavelets in order to perform a statistical analysis. In this case, the mean P-wave quality factor of the ice (and standard deviation) is $Q_{\text{P}} = 320 \pm 60$. Using the same procedure, we also analyzed the reflected S waves (Fig. 3b), obtaining an average ice quality factor of $Q_{\text{S}} = 205 \pm 50$. These values agree with the result provided by the diving and refracted waves. However, this analysis

shows a slight decrease in ice quality factors compared to the maximum values estimated at the bottom of the firn. This decrease reflects the substantial dependence of the quality factor on temperature (e.g., Kuroiwa, 1964; Peters et al., 2012), which increases with respect to depth in polar ice sheets and ice streams. In our case, at the SLW site, the temperature increases from a minimum value of about $-25\,°C$ (annual mean temperature) near the surface to a maximum value at the bed which is close to its pressure melting point (Engelhardt and Kamb, 1993; Tulaczyk et al., 2014).

## 6 Conclusions

A physical explanation of the seismic attenuation (quality factor) in the polar snow and ice masses is essential to gaining insight into the ice sheet and deeper geological formations. This study shows a novel approach, based on a physical model which depends on parameters that can be derived from core sample analyses, to interpret and explain the observed seismic wave attenuation in firn and in the underlying ice. In particular, the proposed model is useful for performing data processing and amplitude variations with offset inversions (AVOs) to extract basal petrophysical properties from active-source multichannel seismic data.

In this work, we estimated the P- and S-wave attenuation profiles in the firn and underlying ice of Whillans Ice Stream (West Antarctica) from the spectral analysis of diving, refracted, and reflected waves of active-source three-component seismic data. The resulting experimental quality factors range from values lower than 5 at the surface to approximately 300 and 250 at about 60 m depth for P and S waves, respectively. Then, the P-wave quality factor further increases up to a maximum value of about 380 in the underlying ice. This attenuation model allowed us to infer the average P- and S-wave quality factors of the entire ice column beneath the firn up to the bed, which is otherwise not possible using standard methods. The estimated average $Q$ factors are slightly lower than the maximum values for the ice at the firn bottom, i.e., approximately 320 and 205 for P and S waves, respectively. This decrease is in agreement with the increase in temperature as a function of depth typical of the polar ice caps.

The firn wave propagation model combines White's mesoscopic attenuation theory of interlayer flow with that of Biot/squirt flow, and the fluid saturating the pores is assumed to be fluidized snow. The dominant attenuation mechanism is found to be that of the global Biot flow, with the theory showing good agreement with the experimental values for the whole firn column. The model also predicts that below the pore close-off depth (approximately 35 m in this case), the squirt-flow loss predominates over the global Biot loss. The proposed model enables seismic experiments to be interconnected with mechanical laboratory analyses of firn samples collected on a study site, allowing for both the calibration of seismic surveys and the inversion of firn properties. To achieve this aim, knowledge of both the seismic velocity and the attenuation is also important because they could allow, at least theoretically, the porosity (and density) profile of the polar firn to be obtained by means of surface or borehole seismic experiments, which is useful for estimating the ice-sheet mass balance from satellite observations of ice-sheet elevation. In light of this information, surface seismic experiments or well logging in inexpensive hot-water-drilled holes could replace costly core drills.

## Appendix A: Mesoscopic loss model

The complex P-wave modulus of two periodic thin porous layers is (White et al., 1975; Carcione, 2022)

$$E = \left[ \frac{1}{E_0} + \frac{2(r_2 - r_1)^2}{i\omega(d_1 + d_2)(I_1 + I_2)} \right]^{-1}, \tag{A1}$$

where

$$E_0 = \left( \frac{p_1}{E_{G_1}} + \frac{p_2}{E_{G_2}} \right)^{-1}, \tag{A2}$$

with $p_l = d_l/(d_1 + d_2)$, $l = 1, 2$,

$$r = \frac{\alpha M}{E_G}, \tag{A3}$$

$$I = \frac{\eta}{\kappa a} \coth\left( \frac{ad}{2} \right), \quad a = \sqrt{\frac{i\omega \eta E_G}{\kappa M E_m}}, \tag{A4}$$

where $\omega$ is the angular frequency, and $i = \sqrt{-1}$.

For each layer,

$$E_m = K_m + \frac{4}{3}\mu, \tag{A5}$$

is the dry-rock P-wave modulus, where $K_m$ and $\mu$ are, respectively, the bulk and shear moduli,

$$E_G = E_m + \alpha^2 M, \tag{A6}$$

is Gassmann's P-wave modulus,

$$K_G = K_m + \alpha^2 M, \tag{A7}$$

is Gassmann's bulk modulus,

$$\alpha = 1 - \frac{K_m}{K_s}, \quad \text{and} \quad M = \left( \frac{\alpha - \phi}{K_s} + \frac{\phi}{K_f} \right)^{-1}, \tag{A8}$$

where $\phi$, $K_s$, and $K_f$ denote the porosity and the bulk moduli of the grains and saturant fluid, respectively. The coefficient $\alpha$ is known as the effective stress coefficient of the bulk material. Finally, $\eta$ is the fluid viscosity and $\kappa$ is the frame permeability.

Let $\rho_s$ and $\rho_f$ denote the mass densities of the grains and fluid, respectively, and let

$$\rho = (1 - \phi)\rho_s + \phi\rho_f, \tag{A9}$$

denote the mass density of the bulk material.

The complex velocity is

$$v_1 = \sqrt{\frac{E}{\rho}}. \tag{A10}$$

This model considers only the P-wave attenuation, since the shear modulus $\mu$ is real.

## Appendix B: Biot/squirt-flow model

The squirt-flow poroelasticity stiffnesses are the Gassmann bulk and shear moduli,

$$K_G = K_m + \alpha(K_m)^2 M(K_m) \ \text{ and } \ \mu_G = \mu_m , \tag{B1}$$

where

$$\alpha(K_m) = 1 - \frac{K_m}{K_s} \ \text{ and }$$

$$M(K_m) = \frac{K_s}{1 - \phi - K_m/K_s + \phi K_s/K_f}, \tag{B2}$$

$\phi$ is the porosity, $K_m$ and $\mu_m$ are the bulk and shear moduli of the drained matrix, and $K_s$ and $K_f$ are the solid and fluid bulk moduli, respectively. We explicitly indicate the functional dependence of $\alpha$ and $M$ on $K_m$, since we shall replace this modulus by a modified matrix (or frame) complex modulus $\overline{K}$ which includes the squirt-flow mechanism. In the same manner, $\mu_m$ will be replaced by $\overline{\mu}$. The new moduli are complex valued and frequency dependent.

In the Biot/squirt-flow model, the bulk and shear moduli of the saturated rock at low frequencies are given by Gassmann's equations,

$$K_G = \overline{K} + \alpha^2(\overline{K})M(\overline{K}) \ \text{ and } \ \mu_G = \overline{\mu}, \tag{B3}$$

where $\overline{K}$ and $\overline{\mu}$ are the bulk and shear moduli of the modified frame, including the unrelaxation due to the presence of the squirt-flow mechanism, and $\alpha$ and $M$ are given by Eq. (B2) with $K_m$ substituted by $\overline{K}$. For simplicity, we keep the same notation for the Gassmann moduli, but now they are complex valued and frequency dependent.

Gurevich et al. (2010) obtained the modified dry moduli in the following form:

$$\frac{1}{\overline{K}} = \frac{1}{K_h} + \left[ \left( \frac{1}{K_m} - \frac{1}{K_h} \right)^{-1} + \left( \frac{1}{K_f^*} - \frac{1}{K_s} \right)^{-1} \phi_c^{-1} \right]^{-1},$$

$$\frac{1}{\overline{\mu}} = \frac{1}{\mu_m} - \frac{4}{15} \left( \frac{1}{K_m} - \frac{1}{\overline{K}} \right) \tag{B4}$$

where $K_m$ and $\mu_m$ are the dry-rock bulk and shear moduli at the confining pressure $p_c$, $K_h$ is the dry-rock bulk modulus at a confining pressure where all the compliant pores are closed, i.e., an hypothetical rock without the soft porosity, and $\phi_c$ is the compliant porosity. This is so small – nearly 0.001 for most rocks – that the total porosity $\phi$ can be assumed to be equal to the stiff porosity. The key quantity in Eqs. (B4) is the effective bulk modulus of the fluid saturating the soft pores:

$$K_f^* = \left[ 1 - \frac{2J_1(kR)}{kRJ_0(kR)} \right] K_f,$$

$$k = \frac{2}{h}\sqrt{-\frac{3i\omega\eta}{K_f}} = \frac{1}{R}\sqrt{\frac{-8K_f^*}{K_f}}, \tag{B5}$$

where $J_0$ and $J_1$ are Bessel functions.

The complex velocities of the P and S waves are

$$v_{2P} = \sqrt{\frac{\overline{K} + 4\overline{\mu}/3}{\rho}}, \tag{B6}$$

and

$$v_{2S} = \sqrt{\frac{\overline{\mu}}{\rho}}, \tag{B7}$$

respectively.

*Data availability.* The data presented in this work are available through the following previous publications: Horgan et al. (2012) and Picotti et al. (2015).

*Author contributions.* SP mainly conceived the study, processed and analyzed all data, produced most of the figures, and wrote the manuscript. SP and MP designed the experiment and acquired the seismic data in Antarctica. JMC contributed to the basic ideas, proposing the theoretical models, producing some of the figures, and writing several parts of the text. All authors discussed the results and were involved in drafting the manuscript.

*Competing interests.* The contact author has declared that none of the authors has any competing interests.

ther geographical representation in this paper. While Copernicus Publications makes every effort to include appropriate place names, the final responsibility lies with the authors.

*Acknowledgements.* We would like to thank Huw J. Horgan and Sridhar Anandakrishnan for their support.

*Financial support.* This work has been supported by the Italian National Program of Antarctic Research (PNRA – WISS-LAKE Project) and by the National Science Foundation (grant no. 0944794, 0632198, 0424589). TS19

*Review statement.* This paper was edited by Adrian Flores Orozco and reviewed by Rolf Sidler and Matthias Steiner.

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

**Remarks from the language copy-editor**

CE1      Is it intended that the citation of Herglotz, Wiechert and Nowak where deleted during your text change?

CE2      Please note that a slash can be ambiguous. Should it be changed to "and", "or" or "and/or"?

**Remarks from the typesetter**

TS1      Please note that units have been changed to exponential format throughout the text. Please check all instances.

TS2      Please give an explanation of why this needs to be changed. We have to ask the handling editor for approval. Thanks.

TS3      Please check the unit Pa s.

TS4      Please give an explanation of why this needs to be changed. We have to ask the handling editor for approval. Thanks.

TS5      Please give an explanation of why this needs to be changed. We have to ask the handling editor for approval. Thanks.

TS6      Please give an explanation of why the figure needs to be changed. We have to ask the handling editor for approval. Thanks.

TS7      Please give an explanation of why this needs to be changed. We have to ask the handling editor for approval. Thanks.

TS8      Please give an explanation of why the equation needs to be changed. We have to ask the handling editor for approval. Thanks.

TS9      Please give an explanation of why this needs to be changed. We have to ask the handling editor for approval. Thanks.

TS10      Please give an explanation of why this needs to be changed. We have to ask the handling editor for approval. Thanks.

TS11      Please give an explanation of why this needs to be changed. We have to ask the handling editor for approval. Thanks.

TS12      Please give an explanation of why this needs to be changed. We have to ask the handling editor for approval. Thanks.

TS13      Please give an explanation of why this needs to be changed. We have to ask the handling editor for approval. Thanks.

TS14      Please give an explanation of why this needs to be changed. We have to ask the handling editor for approval. Thanks.

TS15      Please confirm.

TS16      Please confirm.

TS17      Please confirm.

TS18      Please give an explanation of why this needs to be changed. We have to ask the handling editor for approval. Thanks.

TS19      Please confirm both acknowledgements and financial support sections.