# Peer review of "Seismic attenuation in Antarctic firn"

_The Cryosphere, 2023_

## Referee Comment (RC1)

**REVIEW OF MANUSCRIPT TC-2023-19: SEISMIC ATTENUATION IN ANTARCTIC FIRN**

The study derives phase velocities and attenuation of P- and S-waves in polar firn from seismic experiments using an inversion algorithm. The basic model of the inversion is a stack of homogeneous layers. The attenuation of each layer is obtained from the phase shift between the reference wavelet and the first break wavelet at the corresponding offset. The reference e.g. source wavelet is obtained from receivers placed close to the source.

Parameters for the Biot model are derived from the density model using functions of the porosity that have been shown to be functional in snow.

The study shows, that wave velocities in firn can not be explained by a basic porous model consisting of a rigid ice structure and a pore space consisting simply of air. Such a basic ice-air model underestimates the P-wave attenuation by at least two orders of magnitude.

The authors therefore use a porous model with the pore space filled with so called 'fluidized snow'. Fluidized snow is a mixture of snow particles and air. The difference between the soft air and the relatively rigid snow leads to higher attenuation as of higher friction of the rigid phase and the larger slip due to the soft phase being squeezed. The mechanism and the orders of magnitude of the attenuation is similar to a pore fluid consisting of air and water in melting snow. To my knowledge, the concept of fluidized snow as a pore fluid in Biot-theory to express wave velocity and attenuation in firn is new and has not been applied before this study.

As described in the introduction of the manuscript, only little information of wave velocity and attenuation of firn is present today. Such information is important as it could be used, for example, to obtain porosity of firn with seismic borehole experiments. As such, borehole logging in cheap hot water drill holes could replace costly core drills. Knowledge of the wave attenuation in the firn layer of ice sheets is also important to seismic surveys obtaining geological information of the underlying geological structures.

**1. COMMENTS**

**Figure Attenuation versus density:** In the manuscript the attenuation and velocities are only shown versus depth in the firn deposits. For the use in further investigation and to compare with alternative theoretical models it is crucial to have the velocities and attenuation as a function of density and/or porosity. I would therefore highly recommend to add such figures to the manuscript.

**Physical model of fluidized snow:** For the theoretical model, fluidized snow is assumed to fill half the available pore space. Where the available pore space is a function of density. Given the results of the study this is a good approximation. It is, however, not so clear, how the fluidized snow phase can be measured in firn samples as for example in drilled ice cores. This is not a flaw of the study, but rather a question, that arrises from the results of the study.
* * *
*Date*: March 5, 2023.

**2. Technical corrections**

**Technical corrections:**

- Line 250: "depth depth"

---

## Referee Comment (RC2)

[referee-annotated manuscript omitted]

---

## Author Comment (AC1)

**Author responses to review 1 of: Seismic attenuation in Antarctic firn**

Stefano Picotti[1], José M. Carcione[1,2], and Mauro Pavan[3]

[1]National Institute of Oceanography and Applied Geophysics (OGS), Trieste, Italy.
[2]School of Earth Sciences and Engineering, Hohai University, Nanjing, China.
[3]University of Genova, Italy.

**Correspondence:** Stefano Picotti (spicotti@ogs.it)

**1  Introduction**

First of all, we would like to thank the reviewer for the valuable and constructive comments on our manuscript!
Reviewer's comments are indicated in italic font in this letter. Moreover, new and revised paragraphs which were included in the main manuscript are indicated by the respective line numbers.

According to the points raised below, we made some improvements to the computation of velocities and quality factors (sections 4.1 and 4.2). Furthermore, we added some paragraphs to the Introduction and Conclusions, to better explain the main purposes of our paper and possible new applications of the described menthodology.

**2  Reviewer 1**

1. *The study derives phase velocities and attenuation of P- and S-waves in polar firn from seismic experiments using an inversion algorithm. The basic model of the inversion is a stack of homogeneous layers. The attenuation of each layer is obtained from the phase shift between the reference wavelet and the first break wavelet at the corresponding offset. The reference e.g. source wavelet is obtained from receivers placed close to the source. Parameters for the Biot model are derived from the density model using functions of the porosity that have been shown to be functional in snow. The study shows, that wave velocities in firn can not be explained by a basic porous model consisting of a rigid ice structure and a pore space consisting simply of air. Such a basic ice-air model underestimates the P-wave attenuation by at least two orders of magnitude. The authors therefore use a porous model with the pore space filled with so called 'fluidized snow'. Fluidized snow is a mixture of snow particles and air. The difference between the soft air and the relatively rigid snow leads to higher attenuation as of higher friction of the rigid phase and the larger slip due to the soft phase being squeezed. The mechanism and the orders of magnitude of the attenuation is similar to a pore fluid consisting of air and water in melting snow. To my knowledge, the concept of fluidized snow as a pore fluid in Biot-theory to express wave velocity and attenuation in firn is new and has not been applied before this study. As described in the introduction of the manuscript, only little information of wave velocity and attenuation of firn is present today. Such information is im-*

*portant as it could be used, for example, to obtain porosity of firn with seismic borehole experiments. As such, borehole logging in cheap hot water drill holes could replace costly core drills. Knowledge of the wave attenuation in the firn layer of ice sheets is also important to seismic surveys obtaining geological information of the underlying geological structures.*

I would like to thank the reviewer for the clear synthesis of our work.

We agree with the reviewer that "seismic attenuation in firn can not be explained by a basic porous model consisting of a rigid ice structure and a pore space consisting simply of air". According to the above reviewer comments, we added the following paragraph at the end of the Introduction (starting from Line 61), in order to better explain the main purposes of our paper:

"In the case of Antarctic firn, attenuation of seismic waves cannot be explained by adopting a simple porous model consisting of a rigid ice structure and a pore space filled with air. Such a basic ice-air model underestimates seismic attenuation by orders of magnitude. Therefore, the fluid saturating the pore space in this case is assumed to be fluidized snow, a mixture of snow particles and air (Mellor, 1974; Maeno and Nishimura, 1979; Nishimura, 1996). In this study we show that replacing air with fluidized snow in the pores leads to higher attenuation, and to quality factor values comparable with those obtained from the seismic data. To our knowledge, this is the first attempt to use the concept of fluidized snow as a pore fluid in Biot-theory to model wave attenuation in firn."

We also added the following paragraph at the end of the Conclusions:

"Knowledge of seismic velocity and attenuation is also important because they could allow, at least theoretically, to obtain the porosity profile of polar firn by means of surface seismic or borehole experiments. Therefore, borehole logging in cheap hot water drill holes could replace costly core drills."

2. *In the manuscript the attenuation and velocities are only shown versus depth in the firn deposits. For the use in further investigation and to compare with alternative theoretical models it is crucial to have the velocities and attenuation as a function of density and/or porosity. I would therefore highly recommend to add such figures to the manuscript.*

We made some improvements to the computation of velocities and quality factors. As explained in Section 3.3 and 4.1, the quality factor profiles obtained using the layer-stripping method, strongly depend on the thickness and seismic properties of the shallowest layer. While these properties are well constrained for the $Q_P$ profile, it was not possible to properly constrain the thickness of the shallowest layer related to the $Q_S$ profile. This is due to the fact that the S-wave velocity close to the surface is too low. Therefore, we revised the S-wave velocity curve computed by Picotti et al. (2015),

obtaining higher velocities in the first 15 m below the surface. We changed the Lines 206-209 as follows:

"Calculating the maximum penetration depth of the ray emerging at 6 m offset, we estimated the thickness of this layer to be about 1.6 m thick. Moreover, the estimated source dominant frequency and variance are $f_S$= 496 Hz and $\sigma_S$= 146 Hz, respectively."

As expected, after these corrections the shape of the $Q_S$ profile changed, showing lower quality factors in the deeper parts of the firn. We changed the Lines 223-224 as follows:

"The second plot (Fig. 9) exhibit a moderate increase of $Q_S$, from the previously computed minimum value of 1.9±0.5 close to the surface, to an average maximum value of about 210±77, which remains almost constant at depths larger than 35 m."

We also changed the Line 283 in the Conclusions as follows:

"The resulting experimental quality factors range from values as low as 5 at the surface to approximately 300 and 200 at about 60 m depth, for P and S-waves respectively."

In order to improve the theoretical model, we modified the assumptions on the Poisson ratio of the layers. We changed the Lines 241-245 as follows:

"The physical properties of the firn layer are derived from the density model using functions of porosity that have been shown to be suitable for snow. The density profile as a function of depth is obtained by using the following empirical relationship

$$\rho(z) = 0.917 \left[ 1 + \left( \frac{V_{P,ice} - V_P(z)}{2250} \right)^{1.22} \right]^{-1}, \tag{1}$$

Kohnen (1972), where $V_P(z)$ is the vertical P-wave velocity displayed in Fig. 4 (a), and $V_{P,ice}$ = 3864 m/s is the ice P-wave velocity, which we assume equal to the maximum computed P-wave velocity.
The porosity obtained from the experimental density (1) is

$$\phi(z) = \frac{\rho_s - \rho(z)}{\rho_s - \rho_f}. \tag{2}$$

Fig. 11 shows the experimental density (a) and porosity (b) as a function of depth. These quantities increase monotonically with depth, mainly due to compaction.

Then, for each layer, the dry-rock bulk modulus which best fits the data of Johnson (1982) is

$$K_m = K_s(1-\phi)^{30.85/(7.76-\phi)}. \tag{3}$$

The dry-rock shear modulus is

$$\mu = \frac{3}{2}\frac{1-2\nu}{1+\nu}K_m, \quad \nu = 0.38 - 0.36\phi, \tag{4}$$

where $\nu$ is the Poisson ratio. For $\phi \leq 6\ \%$ ($\rho \geq 870$ kg/m$^3$) the medium is almost ice, and the Poisson ratio is better approximated from the inverted wave velocities as follows

$$\nu = \frac{V_P{}^2 - 2V_S{}^2}{2\left(V_P{}^2 - V_S{}^2\right)} \approx 0.32, \tag{5}$$

(Mavko et al., 2009), where $V_P$ and $V_S$ are the P- and S-wave velocities displayed in Fig. 4(a), respectively."

We added the following reference to the bibliography:

Mavko, G., T. Mukerji, and J. Dvorkin, 2009, The rock physics handbook: tools for seismic analysis in porous media, Cambridge Univ. Press.

Being these assumptions more realistic, the comparison between the experimental and theoretical quality factors also improved, in particular for the S waves. The new figures 4(a) and 13 are shown below. As requested by the reviewer, we added a second abscissa on the top of each figure, to represent the density corresponding to the depths shown at the bottom abscissa.

[Figure]

**Figure 1.** P-wave and S-wave velocity profiles versus depth, obtained using the Hergloz-Wiechert traveltime inversion method. The velocity curves are also represented versus density, obtained using equation (1), proposed by Kohnen (1972).

[Figure]

**Figure 2.** Comparison between the experimental (symbols) and theoretical (solid lines) P- and S-wave quality factors (a) as a function of depth. The quality factors are also represented versus density, obtained using equation (1), proposed by Kohnen (1972). Experimental errors (b) in the computation of $Q_P$ and $Q_S$ from the layer-stripping frequency-shift method.

105   3. *For the theoretical model, fluidized snow is assumed to fill half the available pore space. Where the available pore space is a function of density. Given the results of the study this is a good approximation. It is, however, not so clear, how the fluidized snow phase can be measured in firn samples as for example in drilled ice cores. This is not a flaw of the study, but rather a question, that arrises from the results of the study.*

110

It is important for this study to measure the properties of the fluidized snow phase. This task can be performed in drilled cores of firn samples as indicated in Nishimura (1991). The main apparatus consists of two parts: fluidized snow feeder and inclined chute, where snow can be kept in disintegrated and fluidized conditions. The measurements are carried out at a temperature of $-15\,°C$ to avoid the effects of adhesion between snow particles. In this context, bulk density, elastic
115   velocity and viscosity can be measured.

We added this paragraph at Line 245, and the following reference to the bibliography:

Nishimura, K., 1991, Studies on the fluidized snow dynamics. Contributions from the Institute of Low Temperature Sci-
120   ence, Ser. A, No. 37, pp. 1-55 (Doctor Thesis, Hokkaido University).

4. *Line 250: "depth depth".*

125   Corrected.

---

## Author Comment (AC2)

**Author's responses to the reviewer 2. Seismic attenuation in Antarctic firn**

Stefano Picotti[1], José M. Carcione[1,2], and Mauro Pavan[3]

[1]National Institute of Oceanography and Applied Geophysics (OGS), Trieste, Italy.
[2]School of Earth Sciences and Engineering, Hohai University, Nanjing, China.
[3]University of Genova, Italy.

**Correspondence:** Stefano Picotti (spicotti@ogs.it)

**1  Introduction**

We thank the reviewer for the valuable and constructive comments on our manuscript.

The reviewer's comments are indicated in italics throughout this letter. Also, new and revised paragraphs that have been included in the main manuscript are indicated by their line numbers. Please note that the equation numbers in this letter do not coincide with those in the revised version of the manuscript.

Based on the points raised below, we have made some improvements to the calculation of velocities and quality factors (sections 4.1 and 4.2). Moreover, we added a discussion to explain in detail the importance of this model for studies related to the physical properties of the firn and the characterization of subglacial materials using amplitude variations with offset (AVO) analysis. Furthermore, we have added a few paragraphs to the Introduction and to the Conclusions, to better explain the main purposes of our work and the possible new applications of the described methodology.

1. *The manuscript "Seismic attenuation in Antarctic firn" submitted by Picotti et al. presents a relevant study addressing the attenuation of seismic P- and S-waves in Antarctic firn. In particular, the authors develop a model based on novel combination of the Biot theory with the concept of fluidized snow filling the pore space. The results shown in the manuscript demonstrate the validity of the proposed approach and its importance for investigations of Antarctic subsurface structures and properties using seismic methods. Accordingly, the study is particularly relevant for the readership of this journal, and thus should be considered for publication after a thorough revision of the manuscript.*

   We thank the reviewer for the clear summary of our work.

2. *In the current version, the objective is indirectly obvious as from the current literature only sparse information regarding seismic wave velocity and attenuation in firn is available. The introduction should clearly state the objective of this study.*

25    We have modified the paragraph at the beginning of the Introduction (see lines 23 to line 43), to better explain that tools for quantifying the depth dependence of attenuation in ice sheets need to be further developed:

"There are many examples in the literature that used diving waves to estimate the firn velocity-depth structure by picking and inverting first-break traveltimes (e.g., Kirchner and Bentley, 1979; King and Jarvis, 2007; Picotti et al., 2015).
30    The ice-fabric characteristics as a function of depth have been obtained by exploiting the P- and S-wave anisotropic velocities inferred from active-seismic surveys conducted in different settings (e.g., Picotti et al., 2015; Blankenship and Bentley, 1987). However, while tomographic methods for estimating seismic velocity in the whole ice column are well established, algorithms for quantifying the depth dependence of attenuation are underdeveloped. In particular, as far as we know, there are no examples of vertical profiling and modeling of the intrinsic seismic attenuation of P- and S-waves
35    in the polar firn, so far.
Intrinsic loss is often quantified using the inverse quality factor $1/Q$, which represents the fraction of wave energy lost to heat in each wave period (e.g., Carcione, 2022; Gurevich and Carcione, 2022). P-wave quality factors ($Q_P$) in ice have been measured by several authors in various depth ranges, from values as low as 5 in the temperate ice at the surface of mountain glaciers (e.g., Gusmeroli et al., 2010 ) up to 700 within cold polar ice caps (e.g., Bentley and Kohnen, 1976).
40    This wide range of values indicates a strong dependence of the quality factor on temperature, demonstrated by laboratory experiments (e.g., Kuroiwa, 1964). This dependence was also verified by Peters et al. (2012) from seismic measurements in Greenland, where $Q_P$ decreases with depth due to an increase in temperature. In this case, $Q_P$ was measured within narrow depth ranges, using strong basal and englacial reflections. Furthermore, it is common practice to measure the average $Q_P$ over the entire ice column using primary and multiple reflection spectra (e.g., Holland and Anandakrishnan,
45    2009; Booth et al., 2012). Regarding the S-wave quality factor ($Q_S$), Clee et al. (1969) and Carcione et al. (2021) measured $Q_S$ in warm mountain glacier ice, reporting values of about 23 and 12, respectively. To our knowledge, no other measurements of $Q_S$ in glacial ice or firn have been published in the literature."

We added two paragraphs at the end of the Introduction (starting from line 69, ending 87), in order to better explain the
50    main purposes of our paper:

"In the case of the Antarctic firn, the seismic wave attenuation cannot be explained by adopting a simple porous model consisting of a rigid structure of ice and a porous space filled with air. Such a simple ice-air model underestimates the seismic attenuation by orders of magnitude. Therefore, the fluid saturating the pore space in this case is assumed to be
55    fluidized snow, a mixture of snow particles and air (Mellor, 1974; Maeno and Nishimura, 1979; Nishimura, 1996). In this study we demonstrate that the replacement of air with pore-fluidized snow leads to increased attenuation, and to quality

factor values comparable with those obtained from seismic data. To our knowledge, this is the first attempt to use the concept of fluidized snow as a porous fluid in Biot's theory to model the wave attenuation in firn."

"In the discussion, we give a detailed explanation of the importance of this model for studies related to the physical properties of the firn and to the characterization of subglacial media using amplitude variations with offset (AVO) analysis (e.g., Peters et al., 2008, Booth et al., 2012). We describe an alternative procedure for calculating the average $Q_P$ and $Q_S$ of the ice column below the firn, using the reflected waves at the bottom of the glacier and the $Q$ profile of the firn. This procedure can be useful in cases where conventional methods of amplitude preconditioning for AVO analysis are not applicable."

3. *The authors should be more critical about the uncertainty associated with their results especially with respect to the seismic/mechanical properties of the first (shallowest) layer. How does this rather large standard deviation affect the application of the layer-stripping method?*

We have made some improvements to the calculation of velocities and quality factors. As explained in Sections 3.3 and 4.1, the quality factor profiles obtained using the layer stripping method strongly depend on the thickness and seismic properties of the topmost layer. While these properties are well constrained for the $Q_P$ profile, it was difficult to correctly constrain the topmost layer thickness for the $Q_S$ profile. This is because the velocity of the S-wave near the surface was too low. Therefore, we revised the S-wave velocity curve calculated by Picotti et al. (2015), obtaining an improved version. In this new version (Fig.5a), errors of velocity versus depth are represented, for both P and S waves. We have changed the paragraph on lines 245-251:

" The velocity curves shown in Fig. 5a represent an improved version of those originally presented in Picotti et al. (2015). The uncertainties in the velocities are obtained by perturbing the first-break travel times, according to the dominant frequency of each picked wavelets, and then repeating the Herglotz-Wiechert inversion to obtain different velocity distributions as a function of depth. The mean and standard deviation of the obtained distributions are displayed. The maximum velocities of the P and S waves, verified using the first arrivals refracted at distant offsets on the seismograms acquired using the explosive source, are 3864±30 m/s and 1947±25 m/s at 60±5 m of depth, respectively. At short offsets, errors increase due to the steep velocity gradient near the surface."

The new Figure 5a is shown below (Fig. 1). Then, we modified the text in lines 231-234:

"By calculating the maximum penetration depth of the emerging ray at an offset of 6 m, we estimated the thickness of this layer to be about 1.6 m. Furthermore, the estimated dominant frequency and variance of the source are $f_S$= 496 Hz and $\sigma_S$= 146 Hz, respectively."

[Figure]

**Figure 1.** P- and S-wave velocity profiles as a function of depth obtained by using the Hergloz-Wiechert traveltime inversion method, where the error in velocity is represented. The velocity curves are also represented as a function of density using equation (13) proposed by Kohnen (1972).

90  As expected, after these corrections the shape of the $Q_S$ profile changed, showing lower quality factors in the deeper parts of the firn layer. We modified the text in lines 261-264:

"The second graph (Fig. 10) shows a moderate increase of $Q_S$, from the previously calculated minimum value of 1.9$\pm$0.4 near the surface, to an average maximum value of about 250$\pm$90, which remains almost constant at depths greater than 95  about 40 m."

Fig. 10 is not attached, but the $Q_S$ plot is displayed in the Figure 2 below. We also modified the text in line 415-417 in the Conclusions:

100  "The resulting experimental quality factors range from values lower than 5 at the surface to approximately 300 and 250 at about 60 m depth, for P and S waves, respectively."

Regarding the computation of uncertainties, we added two paragraphs at lines 235-240 and 253-258:

105  "The frequency centroids of the first breaks of the diving P and S waves are obtained from the spectra of the selected waveforms, as described in Picotti and Carcione (2006). The amplitude integrals are calculated, for each wavelet, in a

frequency band from zero to the maximum frequency of the signal. Since this high cut-off frequency depends on the signal-to-noise ratio, a statistic is performed to evaluate the dispersion due to noise. The mean values of the obtained distributions of the frequency centroids, for both the first P and S wave breaks, are shown in Fig. 8. The corresponding standard deviations are less than 3 Hz."

"The two frequency curves shown in Fig. 8, the velocity profiles, the characteristics of the spectral source and $Q_P$ and $Q_S$ of the most superficial layer, are the inputs of the layer-stripping procedure, to calculate the P- and S-wave quality factor profiles of the entire firn column. The uncertainties in $Q_P$ and $Q_S$ profiles are obtained by repeating the procedure using the previously calculated frequency centroid and velocity distributions and perturbing the quality factors of the more superficial layers by their standard deviations. Then, from the two $Q$-factor distributions in output, we have derived the corresponding profiles of the mean and standard deviation with respect to depth."

In order to improve the theoretical model, we modified the assumptions on the Poisson ratio of the layers. We modified the text in lines 289-306, adding the corresponding new cites:

"The physical properties of the firn layer are obtained from the density model using functions of porosity that have been shown to be suitable for snow. The density profile as a function of depth is obtained by using the following empirical relationship (Kohnen, 1972):

$$\rho(z) = 0.917 \left[ 1 + \left( \frac{V_{P,ice} - V_P(z)}{2250} \right)^{1.22} \right]^{-1}, \tag{1}$$

where $V_P(z)$ is the vertical P-wave velocity displayed in Fig. 5a, and $V_{P,ice}$ = 3864 m/s is the velocity in ice, which we assume equal to the maximum computed P-wave velocity.

The porosity obtained from the experimental density (1) is

$$\phi(z) = \frac{\rho_s - \rho(z)}{\rho_s - \rho_f}. \tag{2}$$

Fig. 12 shows the experimental density and porosity, where the former increase and the latter decrease monotonically with depth, mainly due to compaction. Then, for each layer, the dry-rock bulk modulus which best fits the data of Johnson (1982) is

$$K_m = K_s (1 - \phi)^{30.85/(7.76 - \phi)}. \tag{3}$$

The dry-rock shear modulus is

$$\mu = \frac{3}{2} \frac{1 - 2\nu}{1 + \nu} K_m, \quad \nu = 0.38 - 0.36\phi, \tag{4}$$

where $\nu$ is the Poisson ratio. For $\phi \leq 9$ % ($\rho \geq 850$ kg/m$^3$) the medium is mainly ice, and the Poisson ratio is better approximated from the inverted wave velocities as follows

$$\nu = \frac{V_P{}^2 - 2V_S{}^2}{2\left(V_P{}^2 - V_S{}^2\right)} \approx 0.32, \tag{5}$$

(Mavko et al., 2009), where $V_P$ and $V_S$ are the P- and S-wave velocities displayed in Fig. 5a, respectively."

Being these assumptions more realistic, the comparison between the experimental and theoretical quality factors also improved, in particular for the S waves (see new Fig. 13, now Fig. 14). The new figure 14 is shown below (Fig.2):

4. *These values were obtained as the average and the standard deviation of the Q factors obtained for traces recorded at 7, 8, 9 and 10 m offset from the source (point)? How does this rather large standard deviation affect the application of the layer-stripping method?*

As explained above, we made some improvements to the computation of velocities and quality factors (and corresponding errors). In the new version (lines 224-225 and 231-232) it is now specified that, together with the average of the quality factors of the shallowest layer, we computed also the standard deviations, both for $Q_P$ and $Q_S$. In the old version it was indicated an incorrect value of the error for $Q_P$. In the new version we indicated the correct value: $Q_P = 4.1 \pm 1.3$. The effects of these errors on the application of the layer-stripping method, both for P and S waves, are described in point 3 above.

5. *In the current version, the manuscript does not provide a detailed discussion of the obtained results as reflected by the manuscript structure, which does not include a Discussion section yet solely a section presenting the results.*

A Discussion section has been included (Lines 326-408):

[revised manuscript text omitted]

To support the discussion, new data have been included in the paper. These data are displayed in the new Fig. 3. This plot is shown below (Fig.6) and discussed in the text in lines 110-118:

"All the other data (Fig. 1), aimed at defining the image and anisotropy of the entire ice column, were generated using 0.4 kg PETN (pentaerythritol tetranitrate) charges buried at a depth of 27 m using a hot water drill. Data were recorded on two 48-channel seismic systems and the sensors consisted of alternating single vertical 28 Hz geophones and georods spaced 20 m apart (Fig. 3a). In addition, multi-component data was acquired using three-component continuous recording stations (3C) (Fig. 3b). The 3C stations were first placed along the longitudinal profile, spaced 24 m apart and then moved to the transverse profiles, where the distance was 24 m and 240 m. Each of these stations consisted of a Guralp 40-T broadband seismometer with 40 s corner period, coupled to a Reftek RT-130 data acquisition system equipped with GPS timing. The maximum offset was 4320 m for data recorded using 3C stations and 1910 m for the other data. These data are described in more detail by Horgan et al. (2012) and Picotti et al. (2015)."

6. *The conclusion is a mere summary of the main points of the manuscript, yet it does not interpret the main findings in a broader sense and does not relate them to the objectives stated in the Introduction.*

We added more details to the Conclusions, and two paragraphs in lines 417-421 and 426-430:

"The resulting experimental quality factors range from values lower than 5 at the surface to approximately 300 and 250 at about 60 m depth, for P and S waves, respectively. Thus, the P wave quality factor further increases up to a maximum value of about 380 in the underlying ice. This attenuation model allowed us to infer the average quality factors of the P and S waves of the entire ice column beneath the firn, up to the bed, which is otherwise not possible using standard methods. The estimated average $Q$ factors are slightly lower than the maximum values at the firn bottom, in agreement with the increase in temperature as a function of depth typical of the polar ice caps."

"The knowledge of both the seismic velocity and the attenuation is also important because they could allow, at least theoretically, to obtain the porosity (and density) profile of the polar firn by means of surface or borehole seismic experiments, useful for estimating the ice-sheet mass balance from satellite observations of ice-sheet elevation. In light

[Figure]

**Figure 6.** Explosive-charge seismogram recorded by the vertically-oriented geophones and georods along the longitudinal profile (a). Horizontal transverse components showing reflected SH waves from the ice bottom recorded by the 3C stations (b). A 10-400 Hz band-pass filter is applied to remove the surface waves. The wavelets refracted at the bottom of the firn layer and reflected at the ice-sediments interface are indicated.

of this information, surface seismic experiments or well logging in inexpensive hot-water drilled holes could replace
265    costly core drills."

7. *Throughout the manuscript the authors use rather qualitative formulations to describe their results (e.g., "very low/high"). Considering the strong mathematical and physical background of this study such formulations should be avoided by providing a more quantitative interpretation of results or presentation of findings/values reported in the existing literature. Further (more detailed) comments and suggested (technical) corrections can be found in the annotated manuscript file*
270    *attached here.*

We accepted most comments and suggested corrections (technical) found in the annotated manuscript. Many qualitative formulations have been eliminated or replaced with quantitative formulations. We would like to point out that many of these expressions were already supported by quantitative explanations.

275

8. *Lines 235–240: The grains (ice) have the properties $K_s$ = 10 GPa, $\mu_s$ = 5 GPa (shear modulus) and $\rho_s$ = 917 kg/m³ in both layers. The fluid saturating the pores is assumed to be fluidized snow, which is defined as a mixture of snow particles and air, like powder, having zero rigidity modulus. We consider $K_f$ = 571 MPa, $\rho_f$ = 200 kg/m³ and $\eta$ = 0.1 Pa s. Reference for these values?*

280     *Lines 250–261: The squirt-flow model has the following values of the parameters: $h/R$ = 0.015, $\phi_c$ = 0.0002, $K_h$ = 1.38$K_m$, where $K_m$ is the bulk modulus with the grain contacts and cracks open, $C$ = 0.012 kg$^2$/m$^4$*

Regarding the grain (ice) properties, we cite Gurevich and Carcione (2023) (line 281).

Regarding the squirt-flow model properties, we cite Carcione and Gurevich (2011) (line 310).

285     Regarding the properties of fluidized snow ($K_f$, $\rho_f$ and $\eta$), there is already a sentence specifying the references in line 283.

We added a paragraph (lines 285–290), and respective references.

"For this study it is important to measure the properties of the fluidized phase of the snow. This task can be performed in
290     coring of firn samples as indicated in Nishimura (1991). The main apparatus consists of two parts: fluidized snow feeder and inclined chute, where it is possible to store the disintegrated snow in fluidized conditions. Measurements are made at a temperature of $-15$ °C to avoid adhesion effects between snow particles. In this context, bulk density, elastic velocity and viscosity can be measured."

9. *Line 264: "softer layer with much higher porosity". What is "softer" and "much higher", respectively?*

295

At the beginning of the section we state that "Firn is assumed to be a deposition of two porous media, one snow-like layer with high porosity and the other ice-like with low porosity". Therefore, the term "softer" refers to the layer with higher porosity.

300

10. *Line 266–267: "strong peak", a strong peak in what? "lower frequencies", quantify.*

We specified "strong relaxation peak (high attenuation)..." (line 317).

305

11. *Line 280: "amplitude-versus-offset (AVO)". Why was this not mentioned before?*

As indicated in point 2 above, we point out the importance of this study for AVO in a specific paragraph at the end of
310     the introduction. Also, in the new version of the paper we have added a discussion detailing the utility of this model for AVO inversion.

---

## Author Response (AR1)

**Author's responses to the reviewers 1 & 2.**
**Seismic attenuation in Antarctic firn**

Stefano Picotti[1], José M. Carcione[1,2], and Mauro Pavan[3]

[1]National Institute of Oceanography and Applied Geophysics (OGS), Trieste, Italy.
[2]School of Earth Sciences and Engineering, Hohai University, Nanjing, China.
[3]University of Genova, Italy.

**Correspondence:** Stefano Picotti (spicotti@ogs.it)

**1 Introduction**

We thank the reviewers for the valuable and constructive comments on our manuscript. The reviewer's comments are indicated in italics throughout this letter. New and revised paragraphs that have been included in the main manuscript are indicated by their line numbers.

Based on the points raised below, we have made some improvements to the calculation of velocities and quality factors (Sections 4.1 and 4.2). Moreover, we added a Discussion (Sections 5) to explain in detail the importance of this model for studies related to the physical properties of the firn and the characterization of subglacial materials using amplitude variations with offset (AVO) analysis. Furthermore, we have added a few paragraphs to the Introduction and to the Conclusions, to better explain the main purposes of our work and the possible new applications of the described methodology.

**2 Reviewer 1**

1. *The study derives phase velocities and attenuation of P- and S-waves in polar firn from seismic experiments using an inversion algorithm. The basic model of the inversion is a stack of homogeneous layers. The attenuation of each layer is obtained from the phase shift between the reference wavelet and the first break wavelet at the corresponding offset. The reference e.g. source wavelet is obtained from receivers placed close to the source. Parameters for the Biot model are derived from the density model using functions of the porosity that have been shown to be functional in snow. The study shows, that wave velocities in firn can not be explained by a basic porous model consisting of a rigid ice structure and a pore space consisting simply of air. Such a basic ice-air model underestimates the P-wave attenuation by at least two orders of magnitude. The authors therefore use a porous model with the pore space filled with so called 'fluidized snow'. Fluidized snow is a mixture of snow particles and air. The difference between the soft air and the relatively rigid snow leads to higher attenuation as of higher friction of the rigid phase and the larger slip due to the soft phase being squeezed. The mechanism and the orders of magnitude of the attenuation is similar to a pore fluid consisting of air and water in melting snow. To my knowledge, the concept of fluidized snow as a pore fluid in Biot-theory to express wave*

*velocity and attenuation in firn is new and has not been applied before this study. As described in the introduction of the manuscript, only little information of wave velocity and attenuation of firn is present today. Such information is important as it could be used, for example, to obtain porosity of firn with seismic borehole experiments. As such, borehole logging in cheap hot water drill holes could replace costly core drills. Knowledge of the wave attenuation in the firn layer of ice sheets is also important to seismic surveys obtaining geological information of the underlying geological structures.*

We would like to thank the reviewer for the clear synthesis of our work.

We agree with the reviewer that "seismic attenuation in firn can not be explained by a basic porous model consisting of a rigid ice structure and a pore space consisting simply of air". According to the above reviewer comments, we added a paragraph at the end of the Introduction (Lines 69–76) and Conclusions (Lines 429–433), in order to better explain the main purposes of our paper.

2. *In the manuscript the attenuation and velocities are only shown versus depth in the firn deposits. For the use in further investigation and to compare with alternative theoretical models it is crucial to have the velocities and attenuation as a function of density and/or porosity. I would therefore highly recommend to add such figures to the manuscript.*

We made some improvements to the computation of velocities and quality factors. As explained in Sections 3.3 and 4.1, the quality factor profiles obtained using the layer-stripping method, strongly depend on the thickness and seismic properties of the shallowest layer. While these properties are well constrained for the $Q_P$ profile, it was not possible to properly constrain the thickness of the shallowest layer related to the $Q_S$ profile. This is due to the fact that the S-wave velocity close to the surface is too low. Therefore, we revised the S-wave velocity curve computed by Picotti et al. (2015), obtaining higher velocities in the first 15 m below the surface. In the new version of Fig. 4a (now Fig.5a), errors of velocity versus depth are also represented, for both P and S waves. We changed the Lines 230–233 as follows:

"By calculating the maximum penetration depth of the emerging ray at an offset of 6 m, we estimated the thickness of this layer to be about 1.6 m. Furthermore, the estimated dominant frequency and variance of the source are $f_S$= 496 Hz and $\sigma_S$= 146 Hz, respectively."

As expected, after these corrections the shape of the $Q_S$ profile changed, showing lower quality factors in the deeper parts of the firn. We changed the Lines 262–264 as follows:

"The second graph (Fig. 10) shows a moderate increase of $Q_S$, from the previously calculated minimum value of 1.9±0.4 near the surface, to an average maximum value of about 250±90, which remains almost constant at depths greater than about 40 m."

We also changed the Lines 419–420 in the Conclusions as follows:

"The resulting experimental quality factors range from values lower than 5 at the surface to approximately 300 and 250 at about 60 m depth, for P and S waves, respectively."

In order to improve the theoretical model below the pore close-off depth, we modified the assumptions on the Poisson ratio (Lines 304–307). We reorganized the paragraphs between Lines 285–311.

We added the following reference to the bibliography:

Mavko, G., T. Mukerji, and J. Dvorkin, 2009, The rock physics handbook: tools for seismic analysis in porous media, Cambridge Univ. Press.

Being these assumptions more realistic, the comparison between the experimental and theoretical quality factors improved, in particular for the S waves (see Fig. 14). As requested by the reviewer, we added a second abscissa on the top of Fig. 5a and Fig. 14, to represent the density corresponding to the depths shown at the bottom abscissa.

3. *For the theoretical model, fluidized snow is assumed to fill half the available pore space. Where the available pore space is a function of density. Given the results of the study this is a good approximation. It is, however, not so clear, how the fluidized snow phase can be measured in firn samples as for example in drilled ice cores. This is not a flaw of the study, but rather a question, that arrises from the results of the study.*

We added at Lines 285–289 the following paragraph:

"For this study it is important to measure the properties of the fluidized phase of the snow. This task can be performed in coring of firn samples as indicated in Nishimura (1991). The main apparatus consists of two parts: fluidized snow feeder and inclined chute, where it is possible to store the disintegrated snow in fluidized conditions. Measurements are made at a temperature of $-15\,^{\circ}\mathrm{C}$ to avoid adhesion effects between snow particles. In this context, bulk density, elastic velocity and viscosity can be measured."

We also added the following reference to the bibliography:

Nishimura, K., 1991, Studies on the fluidized snow dynamics. Contributions from the Institute of Low Temperature Science, Ser. A, No. 37, pp. 1–55 (Doctor Thesis, Hokkaido University).

**3 Reviewer 2**

1. *The manuscript "Seismic attenuation in Antarctic firn" submitted by Picotti et al. presents a relevant study addressing the attenuation of seismic P- and S-waves in Antarctic firn. In particular, the authors develop a model based on novel combination of the Biot theory with the concept of fluidized snow filling the pore space. The results shown in the manuscript demonstrate the validity of the proposed approach and its importance for investigations of Antarctic subsurface structures and properties using seismic methods. Accordingly, the study is particularly relevant for the readership of this journal, and thus should be considered for publication after a thorough revision of the manuscript.*

   We thank the reviewer for the clear summary of our work.

2. *In the current version, the objective is indirectly obvious as from the current literature only sparse information regarding seismic wave velocity and attenuation in firn is available. The introduction should clearly state the objective of this study.*

   We have modified a paragraph at the beginning of the Introduction (see Lines 23–42), to better explain that tools for quantifying the depth dependence of attenuation in ice sheets need to be further developed.

   Moreover, we added two paragraphs at the end of the Introduction (Lines 69–76 and 82–87), in order to better explain the main purposes of our paper.

3. *The authors should be more critical about the uncertainty associated with their results especially with respect to the seismic/mechanical properties of the first (shallowest) layer. How does this rather large standard deviation affect the application of the layer-stripping method?*

   We have made some improvements to the calculation of velocities and quality factors. As explained in Sections 3.3 and 4.1, the quality factor profiles obtained using the layer stripping method strongly depend on the thickness and seismic properties of the topmost layer. While these properties are well constrained for the $Q_P$ profile, it was difficult to correctly constrain the topmost layer thickness for the $Q_S$ profile. This is because the velocity of the S-wave near the surface was too low. Therefore, we revised the S-wave velocity curve calculated by Picotti et al. (2015), obtaining an improved version. In this new version (Fig. 5a), errors of velocity versus depth are also represented, for both P and S waves. We have changed the paragraph on lines 244–250, to explain these improvements.

   Then, we modified the text in lines 230–233:

"By calculating the maximum penetration depth of the emerging ray at an offset of 6 m, we estimated the thickness of this layer to be about 1.6 m. Furthermore, the estimated dominant frequency and variance of the source are $f_S$= 496 Hz and $\sigma_S$= 146 Hz, respectively."

As expected, after these corrections the shape of the $Q_S$ profile changed, showing lower quality factors in the deeper parts of the firn. We changed the Lines 262–264 as follows:

"The second graph (Fig. 10) shows a moderate increase of $Q_S$, from the previously calculated minimum value of $1.9\pm0.4$ near the surface, to an average maximum value of about $250\pm90$, which remains almost constant at depths greater than about 40 m."

We also changed the Lines 419–420 in the Conclusions as follows:

"The resulting experimental quality factors range from values lower than 5 at the surface to approximately 300 and 250 at about 60 m depth, for P and S waves, respectively."

Regarding the computation of uncertainties, we added two new paragraphs at lines 234–239 and 251–256.

In order to improve the theoretical model below the pore close-off depth, we modified some assumptions on the Poisson ratio at Lines 304–307. Being these assumptions more realistic, the comparison between the experimental and theoretical quality factors also improved, in particular for the S waves (see new Fig. 13, now Fig. 14). We reorganized the paragraphs between Lines 290–311, adding the corresponding new cites.

4. *These values were obtained as the average and the standard deviation of the Q factors obtained for traces recorded at 7, 8, 9 and 10 m offset from the source (point)? How does this rather large standard deviation affect the application of the layer-stripping method?*

As explained above, we made some improvements to the computation of velocities and quality factors (and corresponding errors). In the new version (lines 223–224 and 230) it is now specified that, together with the average of the quality factors of the shallowest layer, we computed also the standard deviations, both for $Q_P$ and $Q_S$. In the old version it was indicated an incorrect value of the error for $Q_P$. In the new version we indicated the correct value: $Q_P = 4.1\pm1.3$. The effects of these errors on the application of the layer-stripping method, both for P and S waves, are described in point 3 above.

5. *In the current version, the manuscript does not provide a detailed discussion of the obtained results as reflected by the manuscript structure, which does not include a Discussion section yet solely a section presenting the results.*

A Discussion section has been included (Lines 330–411), to explain in detail the importance of this model for studies related to the physical properties of the firn and the characterization of subglacial materials using amplitude variations with offset (AVO) analysis.

To support the discussion, new data have been included in the paper. These data are displayed in the new Fig. 3. This plot is described in the manuscript at Lines 110–118.

6. *The conclusion is a mere summary of the main points of the manuscript, yet it does not interpret the main findings in a broader sense and does not relate them to the objectives stated in the Introduction.*

We added more details to the Conclusions, and two new paragraphs in lines 421–424 and 429–433.

7. *Throughout the manuscript the authors use rather qualitative formulations to describe their results (e.g., "very low/high"). Considering the strong mathematical and physical background of this study such formulations should be avoided by providing a more quantitative interpretation of results or presentation of findings/values reported in the existing literature. Further (more detailed) comments and suggested (technical) corrections can be found in the annotated manuscript file attached here.*

We accepted most comments and suggested corrections (technical) found in the annotated manuscript. Many qualitative formulations have been eliminated or replaced with quantitative ones. We would like to point out that many of these expressions were already supported by quantitative explanations.

8. *Lines 235–240: The grains (ice) have the properties $K_s$ = 10 GPa, $\mu_s$ = 5 GPa (shear modulus) and $\rho_s$ = 917 kg/m$^3$ in both layers. The fluid saturating the pores is assumed to be fluidized snow, which is defined as a mixture of snow particles and air, like powder, having zero rigidity modulus. We consider $K_f$ = 571 MPa, $\rho_f$ = 200 kg/m$^3$ and $\eta$ = 0.1 Pa s. Reference for these values?*
*Lines 250–261: The squirt-flow model has the following values of the parameters: $h/R$ = 0.015, $\phi_c$ = 0.0002, $K_h$ = 1.38$K_m$, where $K_m$ is the bulk modulus with the grain contacts and cracks open, $C$ = 0.012 kg$^2$/m$^4$*

Regarding the grain (ice) properties, we cite Gurevich and Carcione (2023) (Line 281).

Regarding the squirt-flow model properties, we cite Carcione and Gurevich (2011) (Line 314).

Regarding the properties of fluidized snow ($K_f$, $\rho_f$ and $\eta$), there is already a sentence specifying the references in Lines 283–284. Moreover, we added a paragraph (lines 285–289), and a new reference (Nishimura, 1991).

Regarding the constant $C$ in the permeability equation, there are already two references in Line 310: Sidler (2015) and

Gurevich and Carcione (2023).

9. *Line 264: "softer layer with much higher porosity". What is "softer" and "much higher", respectively?*

190    At the beginning of Section 3.2 we state that "Firn is assumed to be a deposition of two porous media, one layer snow-like with high porosity and the other ice-like with low porosity.". Therefore, the term "softer" refers to the layer with higher porosity.

10. *Line 266–267: "strong peak", a strong peak in what? "lower frequencies", quantify.*

195

We specified "strong relaxation peak (high attenuation)..." (Line 319).

11. *Line 280: "amplitude-versus-offset (AVO)". Why was this not mentioned before?*

200    As indicated in point 2 above, we evidenced the importance of this study for AVO in a specific paragraph at the end of the introduction. Moreover, in the new version of the paper we have added a Discussion section detailing the utility of this model for AVO inversion.

---

## Referee Report (RR1)

**REVIEW OF REVISED MANUSCRIPT TC-2023-19: SEISMIC ATTENUATION IN ANTARCTIC FIRN**

The author have made a considerable effort to address the comments of the reviewers. A discussion section has been added, the introduction and the results sections have been extended, new figures were added and existing figures were enhanced. The new figures show additional information on the data used to calculate the attenuation of the firn under observation and to illustrate the influence of the firn layer with its high attenuation on AVO analyses of glacier bed properties. In Figure 14 the density of the firn was added as an alternative axis making it possible to compare the results with porosity based Biot models of snow or firn.

In the introduction and discussion section the authors explain the consequences and benefits of the new findings on the investigation of glacier beds with seismic methods: "Studying the structure of polar ice sheets and basal materials is essential for modeling the response of ice masses to climate change. ... The mechanisms of basal movement of glaciers strongly influence the mass balance of Antarctica and are poorly understood. ... Seismic methods are the only way to map the properties of the glacier beds over large areas."

The study shows a novel approach to interpret and explain the observed seismic wave attenuation in firn. The approach is based on a physical model that uses parameters that can be measured on firn samples in the laboratory by means completely independent from the use of seismic waves. In consequence, the model allows to interconnect seismic experiments with mechanical laboratory analyses of firn samples collected on the study site, allowing for both, calibration of the seismic experiment and inversion for firn properties.

As stated im my previous report, I value the manuscript as innovative, technically sound and important. The revision of the manuscript has further clarified the methods applied, the results obtained and the context which the results impact. I therefore recommend to accept the manuscript in its actual form.

**1. Technical corrections**

**Technical corrections:**

- Line 43: "... not the case for polar firn."
- Line 44: "... about $Q_p = 715$ at ..."
- Line 169: "... method is only applicable ..."
* * *
*Date*: September 28, 2023.

---

## Author Response (AR2)

**Author responses to review 1 of: Seismic attenuation in Antarctic firn**

We would like to thank the reviewer for the valuable and constructive comments on our manuscript. We accepted all the corrections suggested by the reviewer. Moreover, we added two sentences to the Conclusions, based on the useful comments of the reviewer.

Regards,

Stefano Picotti